# Magnetically reshapable 3D multi-electrode arrays of liquid metals for electrophysiological analysis of brain organoids

Enji Kim [1,2,7], Eunseon Jeong [3,7], Yeon-Mi Hong[1,2,7], Inhea Jeong [1,2], Junghoon Kim[2,3], Yong Won Kwon[1,2], Young-Geun Park [1,2], Jiin Lee[3], Suah Choi[3], Ju-Young Kim [2,4], Jae-Hyun Lee [2,4] ✉, Seung-Woo Cho [2,3,4] ✉ & Jang-Ung Park [1,2,4,5,6] ✉

To comprehend the volumetric neural connectivity of a brain organoid, it is crucial to monitor the spatiotemporal electrophysiological signals within the organoid, known as intra-organoid signals. However, previous methods risked damaging the three-dimensional (3D) cytoarchitecture of organoids, either through sectioning or inserting rigid needle-like electrodes. Also, the limited numbers of electrodes in fixed positions with non-adjustable electrode shapes were insufficient for examining the complex neural activity throughout the organoid. Herein, we present a magnetically reshapable 3D multi-electrode array (MEA) using direct printing of liquid metals for electrophysiological analysis of brain organoids. The adaptable distribution and the softness of these printed electrodes facilitate the spatiotemporal recording of intra-organoid signals. Furthermore, the unique capability to reshape these soft electrodes within the organoid using magnetic fields allows a single electrode in the MEA to record from multiple points, effectively increasing the recording site density without the need for additional electrodes.

Brain organoids are 3D neural tissues differentiated from human pluripotent stem cells, such as human induced pluripotent stem cells (hiPSCs), and closely resemble various aspects of the human brain[1–4]. For example, they replicate the developmental stages of the neonatal brain, providing valuable insights into organogenesis[5,6]. Researchers have developed techniques to precisely engineer brain organoids, mimicking both their morphological features and the functional characteristics in specific regions of the brain[7–10]. Moreover, neurological diseases, such as Alzheimer's disease, epilepsy, and Parkinson's disease, can be modeled into organoids, which depict the origin of the disease or abnormal behavior at cellular levels[11–13]. For these reasons, there is substantial interest in these brain organoids because they can offer a promising solution for studying the human brain which, until now, has been restricted due to lack of accessibility to living brains with target diseases or ages. Applications of brain organoids span from fundamental research, such as the evolution of brain or investigations on the origins of specific diseases, to the personalized drug screening. The organoids' unique property of embodying biological and genetic

[1]Department of Materials Science and Engineering, Yonsei University, Seoul 03722, Republic of Korea. [2]Center for Nanomedicine, Institute for Basic Science (IBS), Yonsei University, Seoul 03722, Republic of Korea. [3]Department of Biotechnology, Yonsei University, Seoul 03722, Republic of Korea. [4]Graduate Program of Nano Biomedical Engineering (NanoBME), Advanced Science Institute, Yonsei University, Yonsei, Republic of Korea. [5]Department of Neurosurgery, Yonsei University College of Medicine, Yonsei, Republic of Korea. [6]Yonsei-KIST Convergence Research Institute, Seoul 03722, Republic of Korea. [7]These authors contributed equally: Enji Kim, Eunseon Jeong, Yeon-Mi Hong. ✉e-mail: jhyun_lee@yonsei.ac.kr; seungwoocho@yonsei.ac.kr; jang-ung@yonsei.ac.kr

information from hiPSCs enables investigations of human neural dynamics indirectly but concretely, providing insightful knowledge of electrophysiological activity.

To enhance the practicality of brain organoids for diverse research purposes, characterizing of brain organoids is essential in order to analyze their brain-like features. Conventional biological approaches, such as single-cell RNA sequencing (scRNA-seq) or immunostaining, have been extensively used[14]. Unfortunately, these analyses are restricted to the investigation of the similarity of cellular compositions between organoids and the target brain regions. In addition, such analyses are destructive because sections of the organoids are required for optical observation, and it is impractical to detect the dynamics of neural development due to their limited temporal resolution. Hence, to examine the functional features of brain organoids, collection of the electrical signals that arise from the neurons should be accompanied, leading to the electrophysiological analysis. Specifically, neurons in the brain organoid communicate with each other, and they develop strong neural connections, which allows them to build intricate neural networking circuitry as in other neural tissues[15,16]. Thus, observation of neural networks formed within the brain organoid is essential for exploring the dynamics of their functional networking. For the bioelectronic device to examine the neural networks of organoids, high spatiotemporal resolutions with dense arrays of multiple electrodes must be ensured to capture the rapid changes in the potentials at single-cell levels and to analyze the functional connectivity among neurons[17].

Calcium imaging, which employs fluorescent calcium indicators to monitor neuronal activity of the neurons, has also been utilized in brain organoid studies. However, this optical technique is limited by shallow penetration depth caused by the high scattering properties of brain organoids, and low temporal resolution, which is unsuitable for examining functional connectivity within the brain organoids. Additionally, the conventional electrophysiological analysis of organoids at neural network levels has utilized planar surface-type electrodes without protruding 3D shapes, requiring the sectioning of the organoids in order to provide close contact with the intra-organoidal cells[18]. Although these surface-type electrodes can be used to examine neural communication in a sectioned plane, they limit the detection area only to the particular cross-sectional plane, with impairing the 3D volumetric cytoarchitecture of brain organoids. To overcome these destructiveness and dimensionality limitations of the surface-type electrodes, mechanically flexible designs of electrodes have been suggested[19–24]. The mechanical flexibility of thin, solid-metal electrodes allows them to have external contact with the surface of the organoid, while limiting the source of neural signals only to the surface area. However, brain organoids exhibit a virtue in that the cellular structure and neural networking circuitry are developed in 3D volumetric architectures, which necessitates the examination of the electrophysiological communications throughout the interior volume of the 3D organoids. In other words, the detection of intra-organoid signals, which refers to the electrical signals that arise from the neurons inside the organoids, is imperative to fully understand the functional activity of the organoids. Another strategy to preserve the dimensionality of organoids is the insertion of needle-like 3D probes into the organoids[25,26]. However, these rigid material-based, invasive probes induce mechanical modulus mismatch between the probes and brain organoids, leading to impairment of the function of the brain organoids due to neuronal damage and astrogliosis. The mechanical damage due to the insertion of the rigid electrodes activates the glial cells and disrupts the chronic monitoring of the neural activity of the organoids[27,28]. Thus, the other approach has been suggested to integrate mesh-type electrodes in the early stage of organoid development[29]. Although these meshes use solid-state materials, their stretchable design and fineness are beneficial in that they minimize inflammatory responses, which allows the reliable detection of the neural signals. However, the randomly placed electrodes of these wavering meshes often hinder in-depth analysis because the inconsistent distances between the electrodes preclude the uniform distribution of recording sites within the organoid. Despite these novel approaches, limited control over the number and position of recording sites remains a significant obstacle. Thus, the accurate arrangement of electrode structure and position inside 3D volumetric organoids can be key for the comprehensive spatiotemporal analysis of intra-organoid signals without disrupting the 3D neurological dynamics throughout the brain organoids.

In addition, diversification of the structure design is also important in order to construct MEAs with various geometries according to the individual organoids. For example, the shape and size of brain organoids can be different for each individual, so the locations of multiple electrodes must be different in order to analyze specific regions within the organoid. In this respect, the printing approaches used in the graphic arts, particularly those based on direct writing techniques, are of interest because they provide the flexibility in choices of structure designs so that changes can be made rapidly using software-based printer-control systems.

Herein, we present a magnetically reshapable 3D MEA of liquid metals (LMs) for electrophysiological analysis of brain organoids. The biocompatible liquid metal, i.e., eutectic gallium-indium alloy (EGaIn; 75.5 wt% gallium, 24.5 wt% indium), is printed directly through a nozzle into various heights of 3D pillar shapes to detect intra-organoid signals across diverse 3D coordinates within a brain organoid. Young's modulus of these 3D LM electrodes is comparable to that of the organoid. Their softness and fine protruding structures allow minimal invasion into the organoid, which can be suitable for chronic monitoring of neural activity. This MEA platform includes the following key advantages. First, the placement and height of microscale 3D LM electrodes can be controlled readily by the printing conditions, allowing them to be precisely customized to address the specific 3D morphology of individual organoids. In particular, the high resolution of this 3D LM printing forms cellular-scale dimensions of the tips of the pillars, which are structurally and mechanically similar to neurons in brain organoids while minimizing invasiveness. As a result, the precisely tailored arrangement of the 3D LM electrodes facilitates the monitoring of intra-organoid signals across the entire 3D spatial points of the individual organoid. In addition, the platinum (Pt) nanoclusters only cover the tips of these 3D LM pillars where cell-electrode interfaces are well established, thereby reducing the electrode impedance, increasing their surface area, and improving signal read-out quality. Second, the in-depth analysis of the intra-organoid signals using our 3D LM MEA reveals electrophysiological communications via neural networks developed within the 3D volumetric organoid. In addition, the minimal invasiveness of the 3D LM electrodes permits a tracking maturation of brain organoids at the electrophysiological network level. Last, the intrinsic deformability and the softness of EGaIn facilitate the magnetic control in reshaping these 3D LM electrodes through ferromagnetic layer coating. As an example, the tip of the 3D LM pillars is adjusted according to the direction of the magnetic field, allowing a single electrode to detect multiple areas within the brain organoid, effectively increasing recording site density without the need for additional electrodes. This multi-spot detectable MEA can be advantageous for the 3D mapping of intra-organoid neural networking circuitry, and it offers a promising solution to overcome the limitations of previous methods and providing elaborate insights into the electrophysiological volumetric networks of brain organoids.

## Results
### 3D LM MEA for intra-organoid recording
Figure 1a presents a schematic image on the intra-organoid recording using 3D LM MEA. 3D LM pillars with desired placements and heights were directly printed to form electrodes (Fig. 1b). The printing system

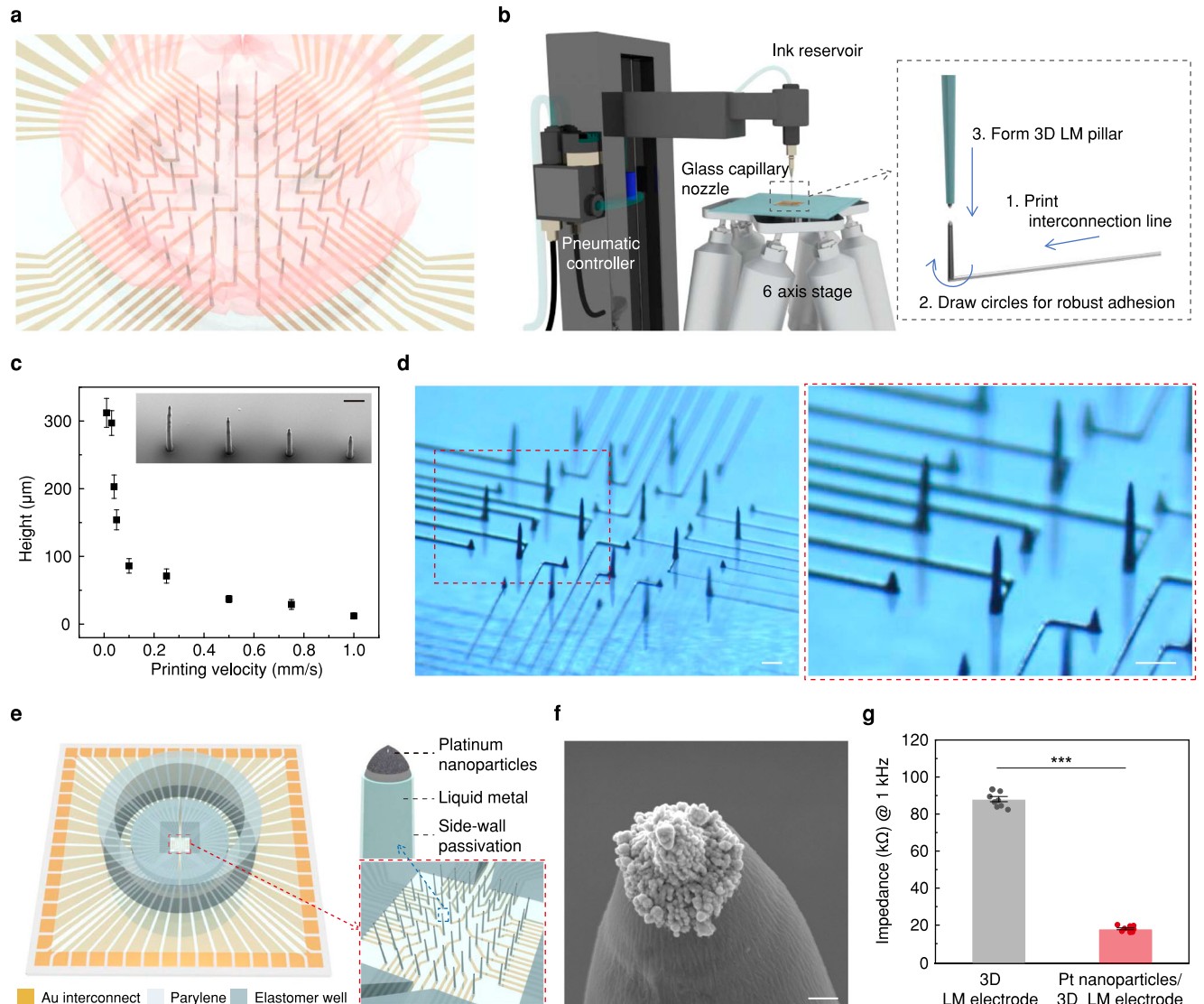

**Fig. 1 | 3D multi-electrode array of liquid metals as an electrophysiological monitoring platform. a** Schematic illustration of intra-organoid signal recording by 3D LM MEA. **b** Schematic illustration of printing system and printing process. **c** Various heights of 3D LM pillars by control printing velocity. Each data point indicates the average of 12 measurements, and the error bars represent the s.e.m. Inset presents SEM image, which was taken by 30° tilting of the substrate, presenting 3D LM pillars with various heights. Scale bar, 100 μm. This experiment was independently repeated more than ten times with similar results. **d** Optical images of LM interconnects and 3D LM pillars with height-variance. Scale bars, 100 μm. **e** Schematic illustrations of 3D LM MEA, and single 3D LM electrode. **f** SEM image of platinum nanoclusters coated on tip of the electrode (The substrate is 30° tilted when the image was taken). Scale bar, 1 μm. This experiment was independently repeated more than ten times with similar results. **g** Comparison of impedances between pristine 3D LM electrode and Pt nanocluster coated LM electrode ($n = 9$ electrodes for each group; $p = 2.27 \times 10^{-13}$). All data are presented as mean ± s.e.m. Statistical differences were determined with unpaired, one-sided t-test; ***$p < 0.001$. LM liquid metal; MEA multi-electrode array; SEM scanning electron microscopy.

was composed of an ink reservoir connected to a glass capillary nozzle in a fixed position, a pneumatic pressure controller, and a precisely controllable 6-axis stage ($x$, $y$, $z$-axis, 2 tilting axes, and a rotating axis in xy-plane). EGaIn, which exists in the liquid phase at room temperature (melting point of 15.5 °C), was used as the ink. The Young's modulus of EGaIn ($2.1 \times 10^5$ Pa) is 4 to 6 orders of magnitude lower than that of the rigid, solid-state metals (e.g., Au or Pt) that are conventionally used for bioelectronics[30,31]. The mechanical softness and good biocompatibility of EGaIn ensures the avoidance of inflammatory responses that can be triggered by the mechanical modulus mismatch between biological tissue and materials[32]. Supplementary Note 1 describes the printing process, and Supplementary Movie 1 shows a movie to print interconnects and 3D pillars. When a pressure of 36 psi was applied to a glass capillary (inner diameter: 18 μm), pillars of varying heights above

310 μm (diameter: 14 μm in minimum) were formed by controlling the printing velocity (Fig. 1c). This controllability over the pillar height allowed precise positioning of the recording sites, which were formed only at the pillar tips (after selectively encapsulating the side walls of pillars using a thin insulating layer), throughout the internal volume of the organoid. For example, Fig. 1d shows photographs of a 3D LM MEA consisting of LM interconnects and pillars of varying heights; these pillars are capable of collecting electrophysiological information across different spaces in the $x$, $y$, and $z$ axes within an organoid (Supplementary Fig. 1).

Figure 1e presents schematic illustrations of our 3D LM MEA and the pillar tip part. This MEA consisted of interconnects, a parylene-passivation layer, 3D LM electrodes (i.e., pillars), and elastomer wells. The elastomer wells included (i) an inner microwell that guided the

positioning of an organoid to the desired location of this MEA device and (ii) an outer macrowell that contained media for the cultivation of organoids. The side walls of 3D LM pillars were encapsulated with parylene-C, a biocompatible elastomeric layer (thickness: 2 μm). Then, using the isotropy of reactive ion etching system, only the tips of pillars were opened selectively, resulting in a small open area of approximately 78.5 μm² (Supplementary Fig. 2). Additionally, the Pt nanoclusters were electrodeposited only on the open tip areas of EGaIn, and the rough surface of these Pt nanoclusters increased the surface area of the pillar tip to establish a good cell-electrode interface (Fig. 1f). For example, Fig. 1g and Supplementary Fig. 3 show that the 3D LM electrode coated with Pt nanoclusters exhibited an impedance that was four times lower than the impedance of the pristine 3D EGaIn electrode (without Pt nanoclusters). Furthermore, when we investigate the stability of the electrical performance of 3D LM electrodes by accelerated aging test, the impedance shows negligible changes for 12 days at 87 °C (18.005 ± 0.083 kΩ to 18.774 ± 0.145 kΩ), which corresponds to 12 months in the incubator (37 °C) (Supplementary Fig. 4).

This direct printing of LM offers flexibility in choice of structural designs, such as 1) variation in the position and height of the 3D electrodes and 2) addressing the diverse needs for achieving electrophysiological information from brain organoids. In particular, an electrophysiological monitoring system for drug screening requires a high-throughput platform to analyze the efficacy of an identical drug to various groups of organoids or to monitor the responses of brain organoids to numerous types of drugs. As an example, Supplementary Fig. 5 demonstrates a high-throughput platform where a total of 9 MEAs (with 3D LM electrodes) were integrated into a single chip to simultaneously monitor the electrophysiological signals from 9 different organoids. This ease of modifying the configurations of the 3D LM electrodes provides new opportunities to explore neural dynamics in brain organoids using customizable designs of devices.

## Integration of 3D LM MEAs with human cortical organoids

Cortical organoids were generated from hiPSCs based on previously established protocols with minor modifications (Fig. 2a)[33]. Briefly, single-cell-dissociated hiPSCs were aggregated with the addition of dual-SMAD inhibition for neural induction. Then, they were further differentiated by the sequential addition of growth factors to the differentiation medium. Early developmental cortical organoids showed an expanded epithelium with bright, smooth, optically translucent edges, and further differentiated organoids increased in size, reaching over 3 mm on day 60 after which little additional growth was observed for up to 120 days of culture (Fig. 2b). The cytoarchitecture of the generated cortical organoids was confirmed by immunofluorescence staining for SOX2, a radial glial cell marker, and TUJ1, a neurofilament protein marker (Fig. 2c). The expression of SOX2 was observed in the cells that lined the ventricular zone, similar to in-vivo development of brain tissue. At 30 days, the majority of ventricular units had strong SOX2+ signals. However, by 120 days, the SOX2+ ventricular zone became ambiguous. After 150 days of culture, we confirmed the presence of glial cells, which occur during the late stages of cortical development in humans, via the expression of glial fibrillary acidic protein (GFAP), an astrocyte marker (Fig. 2d)[34]. The GFAP+ cells, with the characteristic morphological feature of astrocytes, were distributed throughout the organoids. In addition, Vesicular glutamate transporter 1 (VGLUT1)+ mature excitatory neurons were also observed (Fig. 2e).

We then examined whether the cortical organoids exhibited mature cortical organization during long-term culture. To determine if developing cortical organoids could recapitulate the cortical spatial organization, the organoids were stained with markers for cortical layer, i.e., CTIP2 and SATB2, which identify layers V (deep) and II–IV (upper), respectively, on day 60 and 130 of the culture (Fig. 2f, g). The cortical plate of 60-day organoids contained both the deep-layer

marker CTIP2 and the upper-layer marker, SATB2, without layer preference. In contrast, when the cortical organoids were cultured up to day 130, the laminar distribution of the markers became layer-specific. The upper layers were mainly populated by SATB2+ neurons, separated by a distinguishable boundary. We analyzed quantitatively the laminar expression patterns of SATB2 and CTIP2 (Fig. 2g). On day 60, the distributions of the two markers overlapped, but on day 130, they appeared as two separate populations, representing the upper and deep layers. Altogether, these results demonstrate that the generated cortical organoids recapitulate the neurogenesis and formation mature cortical plates that have distinct upper and deep cortical layers with glial cells.

Figure 3a and Supplementary Movie 2 illustrates the integration of this cortical organoid to our 3D LM MEA. After positioning the cortical organoid floating in the medium near the electrode area and using a pipette to aspirate the medium, this organoid settled down. Then, 3D LM electrodes became naturally inserted inside this organoid. After the aspiration for electrode insertion, this organoid recovered its original form by resupplying the medium. Immunostaining of the organoid section confirmed that the 3D LM electrodes had been inserted successfully into the organoid (Fig. 3b). To investigate the biocompatibility of these electrodes on the cortical organoid, a cytotoxicity test using a live/dead staining kit was performed for this sample, and this result was compared with the control case (i.e., organoid with no electrode) for 2 weeks (Fig. 3c). There was no significant difference in viability between these two cases (Fig. 3d). Furthermore, we performed optical clearing of the organoid followed by whole-mount 3D imaging (Supplementary Fig. 6), showing that the 3D LM electrodes were inserted into the organoid straight without causing any structural damage on brain organoids, allowing for reliable monitoring of neural network within the organoid. To further investigate the effect of 3D LM electrodes on neuronal gene expression in the organoid, quantitative real-time polymerase chain reaction (qPCR) was performed on 7 and 14 days after the electrode insertion (Fig. 3e). Expression of neuronal differentiation markers (PAX6, Nestin, TUJ1) showed no significant difference between organoids with or without the 3D LM electrodes. These data suggest that insertion of 3D LM electrodes does not affect the viability or neurogenesis of cortical organoids during the recording period.

## Intra-organoid recordings using 3D LM MEAs

For intra-organoid recordings, 3D volumetric electrophysiological activities of a brain organoid, a 4-month-old cortical organoid that resembles the cerebral cortex of the human brain was monitored using the 3D LM MEA. The intra-organoid signals with frequencies ranging from 0.1 to 3000 Hz were recorded and transferred to a computer passing through a multi-channel signal processor (Tucker-Davis Technologies Inc, USA). The signal processor applied band-pass filters on the data and displayed the results through the software (Synapse). We examined the electrophysiological activities of the organoid by acquisition of filtered data directly from the software. We then utilized custom MATLAB code to gain deeper insight into the 3D neural networking circuitry that was developed within this organoid.

Figure 4a and Supplementary Fig. 7 present the single unit (SU) potentials and local field potentials (LFPs) of this 4-month-old cortical organoid, respectively. The sufficiently low impedance and the cellular-scale size of our 3D LM electrodes allowed the detection of SU potentials, the rapid fluctuations in voltage traces (noise level: ~13 μV). To examine the neuronal activity of intra-organoid signals, we conducted the spike detection using a threshold of -5 × standard deviation of the SU potential. The fluctuations in the voltage traces that exceeded the threshold within 1-2 milliseconds were determined as spikes. Principal component analysis (PCA), which is commonly used to classify clusters of neural spikes, was adopted to examine the spikeforms. The mean spikeforms of each cluster were detected from 60 channels

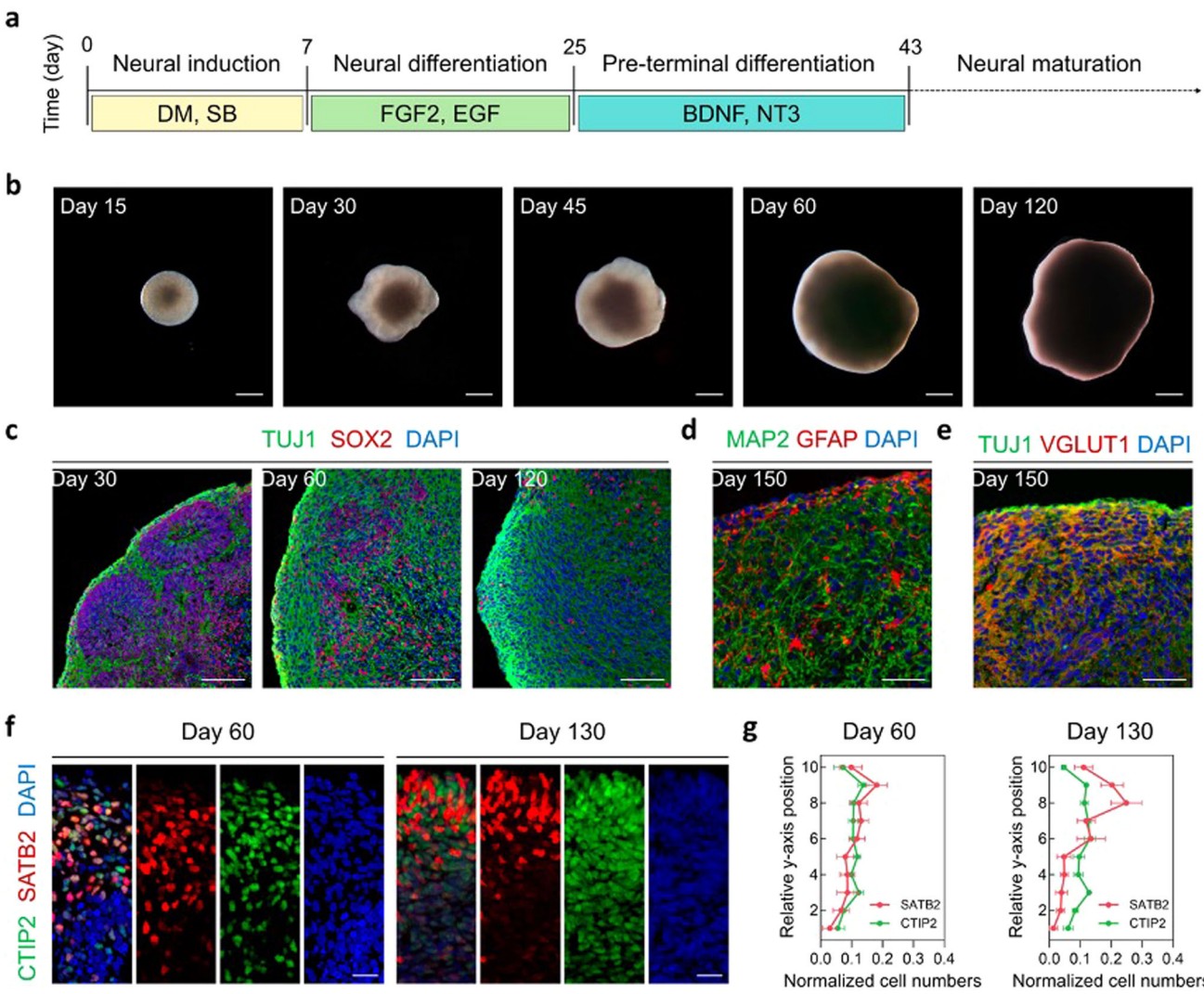

**Fig. 2 | Formation and characterization of cortical organoids. a** Schematic of the protocol for generating human cortical organoids from hiPSCs. DM, Dorsomorphin; SB, SB-431542; FGF2, Fibroblast growth factor 2; EGF, Epidermal growth factor; BDNF, brain-derived neurotrophic factor; NT3, Neurotrophin-3. **b** Representative bright-field images of cortical organoids at 15, 30, 45, 60, and 120 days of culture ($n = 4$-$8$, biological replicates). Scale bars, 500 μm. **c** Immunofluorescence staining for neuronal progenitor marker SOX2 and neuronal marker TUJ1 at 30, 60, and 120 days of culture ($n = 5$, biological replicates). Scale bars, 100 μm. **d** Immunofluorescence staining for neuronal marker MAP2 and astrocyte marker GFAP at day 150 in cortical organoids ($n = 5$, biological replicates).

Scale bar, 50 μm. **e** Immunofluorescence staining for TUJ1 and glutamatergic neuron marker VGLUT1 at day 150 in cortical organoids ($n = 4$, biological replicates). Scale bar, 50 μm. **f** Comparative immunohistochemical images and **g** quantifications of the distribution of SATB2+ and CTIP2+ neurons in the cortical organoids at days 60 and 130. The y-axis position of the images was evenly divided into 10 bins following the basal-to-apical direction within 200 μm from the pial surface. The curve showing the normalized cell number within each bin was calculated by measuring the number of positively stained cells in a bin and dividing them by the total cells. Values were presented as mean ± s.e.m. ($n = 4$, biological replicates). Scale bars, 25 μm. hiPSCs, human induced pluripotent stem cells.

of 3D LM MEA, and representative results are presented in Fig. 4b. Each single electrode detected only two or three clusters, indicating that these 3D electrodes could collect intra-organoid signals without degradation to distinguish differences in spikeforms. We further examined the response of these cortical organoids to potassium chloride (KCl), in order to investigate changes in neural activity. The 10 mM of KCl was added to the culture medium for intra-organoid recordings, and electrophysiological responses were analyzed as shown in Supplementary Fig. 8a. The spiking rate significantly increased by this KCl treatment, indicating that the recorded intra-organoid signals originated from the activity of neurons (Supplementary Fig. 8b). Along with the treatment of KCl, we examined the neural response of brain organoid to tetrodotoxin (TTX) to confirm that the recorded signals originated from neuronal activities (Supplementary Fig. 9).

Brain organoids have attracted researchers' interest because they recapitulate the complex functionality of the human brain. Thus, brain organoids have been exploited in a diverse range of studies, ranging from discovering the dysfunction in neuropathological circuits of specific diseases to excavating patient-specific drugs with patient-driven organoids. The functionality of the brain organoid is represented by communications among neurons through the neural networking circuitry developed in the 3D intra-organoid volume. In other words, simply observing the voltage traces arising from neurons is insufficient to fully understand the connoted electrophysiological information of organoids. Thus, comprehensive examination of 3D intra-organoid neural networking circuitry can exhaustively utilize the merit of brain organoids. In this purpose, we investigated neural networking circuitry based on the evaluation of synchronized activity by calculating spike train synchrony (referred to as synchronization

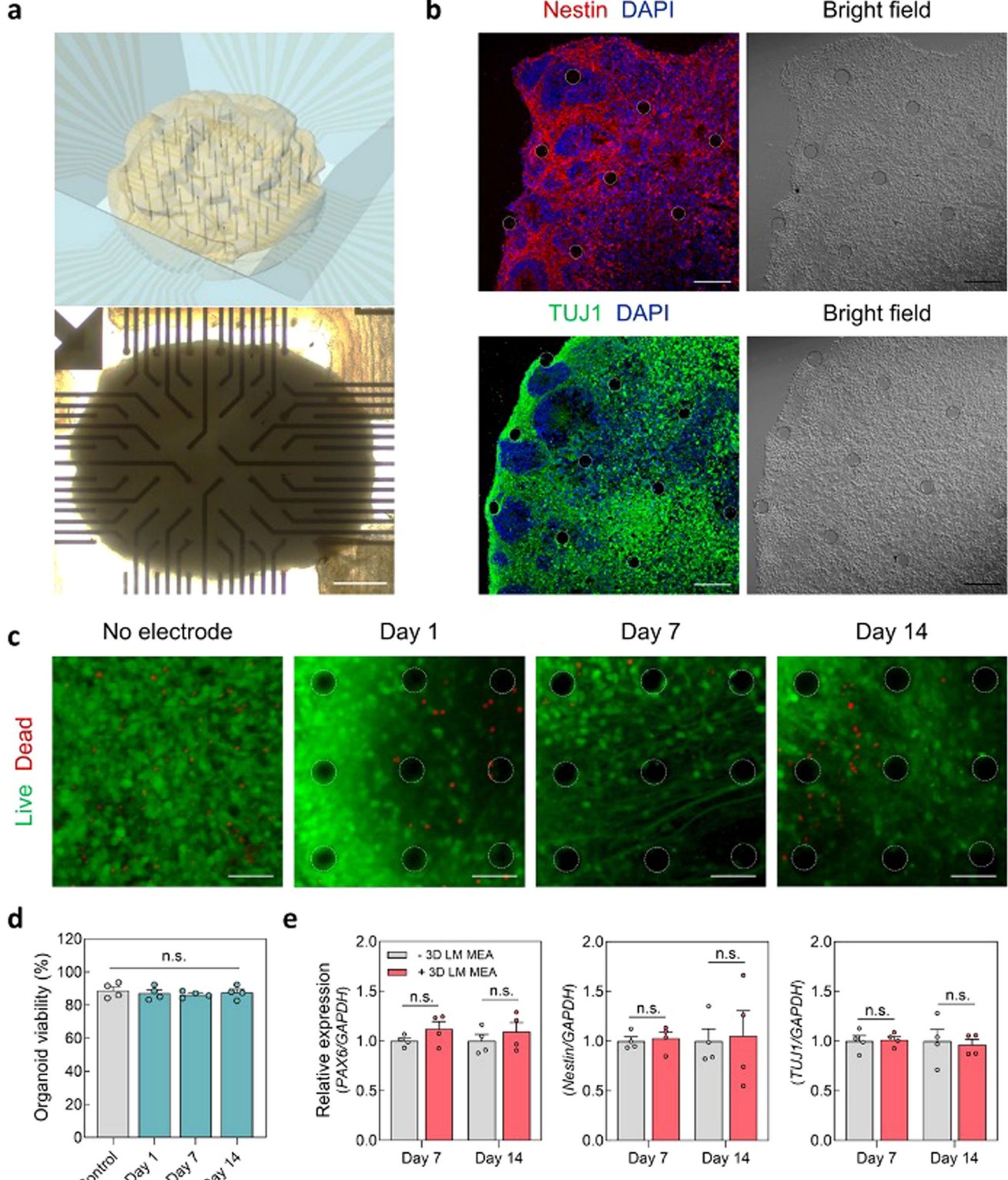

**Fig. 3 | Integration of cortical organoids to 3D LM MEA. a** (top) Schematic illustration and (bottom) representative bright-field image of cortical organoids integrated with 3D LM MEA ($n = 3$, biological replicates). Scale bar, 500 μm. **b** Immunofluorescence images of the sectioned cortical organoids with 3D LM electrodes for (top) Nestin and (bottom) TUJ1. Dashed circles represent the circumference of the 3D LM electrode ($n = 5$, biological replicates). Scale bars, 50 μm. **c, d** Biocompatibility of the 3D LM MEA on cortical organoids; (**c**) live/dead fluorescence images. Scale bars, 50 μm. and (**d**) viability of cortical organoids with or without the 3D LM electrodes at days 1, 7, and 14 of culture after the insertion. Dashed circles represent the circumference of the 3D LM electrode ($n = 4$, biological replicates). **e** The qPCR analysis of cortical organoids with or without 3D LM MEA at days 7 and 14 of culture after the insertion ($n = 4$, biological replicates). All data are presented as mean ± s.e.m. and statistical differences were determined with unpaired, two-sided $t$-test. LM, liquid metal; MEA, multi-electrode array; qPCR, quantitative real-tme polymerase chain reaction.

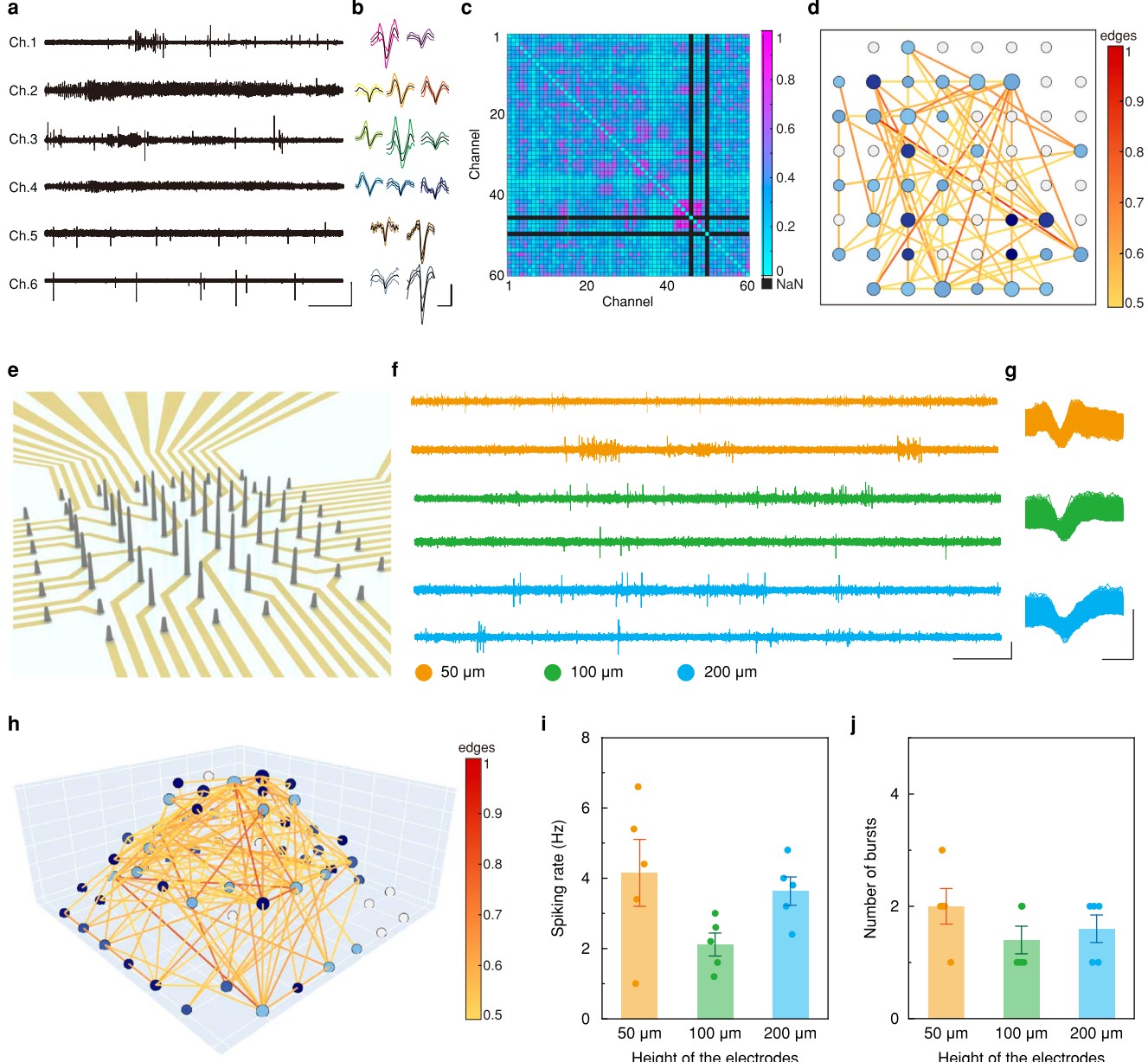

**Fig. 4 | Intra-organoid signals from 4-month-old cortical organoids.**
**a, b** Representative intra-organoid signals of 4-month-old cortical organoid integrated on 3D LM MEA; (**a**) SU potential. Scale bars, 50 μV (vertical), 10 s (horizontal). (**b**) Spikeforms. Scale bars, 50 μV (vertical), 2 ms (horizontal). **c** Color mapping of synchronization score between the electrodes located inside the 4-month-old cortical organoid. The color bar indicates the synchronization score between two electrodes. **d** Neural networking map based on synchronization score and neural community of 4-month-old cortical organoid integrated on 3D LM MEA. Each node indicates individual channels of 3D LM MEA. The lines connecting two electrodes are presented when the synchronization score exceeds 0.5. The size of the node represents the number of lines connecting with other nodes, and the color of the node represents the neural community. The color of lines represents the synchronization score. **e** Configuration of 3D LM electrodes with 3-level of heights. **f, g** Representative intra-organoid signals depending on the heights of the electrodes. Each color represents the height of the electrodes; Orange, 50 μm, green, 100 μm, blue, 200 μm. (**f**) SU potentials. Scale bars, 100 μV (vertical), 0.5 s (horizontal). (**g**) Spikeforms. Scale bars, 100 μV (vertical), 0.5 ms (horizontal). **h** 3D neural networking map corresponding to the 3-level height variance of 3D LM MEA. **i, j** Changes in neural activity depending on the heights of the electrodes ($n = 5$ electrodes per each height); (**i**) Spiking rate, (**j**) Number of bursts. All data are presented as mean ± s.e.m. LM, liquid metal; MEA, multi-electrode array; SU, single-unit.

score[35]). The representative synchronized activities of the brain organoid are presented in Supplementary Fig. 10. The mechanism to determine the synchronization score with this platform is described in Supplementary Note 2. Figure 4c presents the synchronization score mapping of this 4-month-old cortical organoid, providing an intuitive, color-by-color representation of active neural networking. Furthermore, we also performed an evaluation of neural networks by identifying neural communities. We defined a neural community among neurons where a modularity, a complexity of neural networking

circuitry, is maximized[36,37]. (The computational method for neural community is described in Supplementary Note 3) Fig. 4d presents the neural community and connectivity of this 4-month-old organoid as a neural networking map. Nodes were placed according to the distribution of 3D LM electrodes, and the node size indicated its connectivity (i.e., the number of connecting lines with other nodes). Nodes in the same neural community were shown in the same color. As shown by the color bar in Fig. 4d, the line color indicated the synchronization score. These results from neural network mapping indicated that

neurons in contact with electrodes, within a neural community, developed robust and complex circuitry. And the networking circuitry of neurons near electrodes with larger nodes reached diverse regions of this organoid, indicating that the neurons actively communicated with other neuron groups. For example, neurons near the electrodes connected by red lines were connected strongly by neural networking circuitry.

Since the height of 3D LM electrodes can be controlled readily during the direct printing process, the electrodes can be positioned in a variety of configurations throughout the internal volume of organoids to investigate changes in neural activity depending on the recording site. For example, we printed 3D LM pillars with 3 different heights to record SU potentials and spikes at various coordinates within a single organoid (Fig. 4e and Supplementary Fig. 11). Figure 4f, g show the results plotted in different colors depending on the electrode height. The neural network map was also displayed in 3D space corresponding to the configuration of electrodes, representing the neural networks developed in this 3D volumetric organoid (Fig. 4h). In addition, the neural activity of this organoid along the electrode height was analyzed by the parameters of spiking rate and burst number (Fig. 4i, j, and Supplementary Fig. 12). The spiking rate detected at 50- and 200-μm-high electrodes were higher than those detected at 100-μm-high electrodes. On the other hand, the number of bursts, representing the mature activity of the neurons, showed negligible changes with electrode height. These observations indicated that neural activity of this cortical organoid varied longitudinally, while showing an equivalent maturation degree of neurons irrespective of these variances in neural activity. The neural activities from other organoids monitored by identical electrode configuration are shown in Supplementary Figs. 13, 14. Given the variability in cytoarchitecture between brain organoids, the results differ to other organoids. For instance, in Fig. 4i, the spiking rate of the 100 μm-height electrodes is lower than that of the 50 μm and 200 μm height-electrodes, whereas in Supplementary Fig. 13, the spiking rates increase with electrode height. A similar tendency was also observed in the analysis of the intra-organoid signals obtained using another 3D LM MEA with 2 different heights of electrodes (Supplementary Figs. 15–17). The variations in intra-organoid signals corresponding to the electrode heights are consistent with layer-specific expression of discriminative neural markers (Fig. 2f, g) and findings from a previous report[38], which demonstrated distinct electrical properties for each sub-neuronal type constituting the cortical layer. These results indicate that the ability of our 3D LM MEA to detect various 3D coordinates within the organoid facilitates capturing variations in neural activities throughout the organoid, which is inconsistent across different organoids due to the randomly developed cytoarchitectures. Also, these analyses along the electrode height as well as 3D neural networking map provide insight into the neural circuitry between neurons across cortical layers, a capability that has not been achievable with previous approaches using surface-type MEAs.

### Analysis of electrophysiological maturation of cortical organoids

The maturation of intra-neural networks in cortical organoids was investigated using our 3D LM MEA. The minimal invasiveness and organoid-like softness of these electrodes can aid in long-term monitoring of the electrophysiological activities in cortical organoids. Additionally, the high flexibility of electrode configuration (position and height) through direct printing facilitates examination of the intra-neural networks during their maturation, as the electrodes can be tailored to individual organoids that can be mature into a variety of morphologies.

A brain organoid is composed of neurons that generate electrical action potentials, and glial cells that support the neurons and are involved in immune responses. These cells mature as the organoid is cultured and developed. Thus, the neural networking circuitry between cells becomes more complex and stronger. These features of cellular maturation lead to electrophysiological maturation, as presented in Fig. 5 and Supplementary Fig. 18. SU potentials were recorded during the maturation period of this cortical organoid (Fig. 5a), and its electrophysiological properties were analyzed using custom MATLAB codes (Fig. 5b–e). During its maturation from 2 to 6 months, the spiking rate increased (from $0.330 \pm 0.241$ Hz to $8.90 \pm 5.87$ Hz) and inter-spike intervals (ISIs) decreased (from $3.832 \pm 1.60$ s to $0.755 \pm 0.790$ s) as shown in Fig. 5b, c. Furthermore, bursts, repeatedly arising spikes over a minimum duration of 50 ms, were examined during the culture period from 2 to 6 months and both burst number (from $1.5 \pm 1.5$ to $9 \pm 5.209$) and burst duration (from $0.991 \pm 0.165$ s to $1.490 \pm 0.562$ s) increased at the same time (Fig. 5d, e). These results showed the similar trends reported in previous studies[26,39], suggesting that bursts serve as an indirect indicator of the maturation degree in neural networking circuitry. Figure 5f presents neural network maps of 2-, 4-, and 6-month-old organoids, showing the increase in complexity and connectivity among neurons, evidenced by a rising number of lines connecting nodes. Also, as presented in Fig. 5g–j, the changes in properties related to neural networking circuitry were evaluated within the maturation period. The average number of connecting lines per node indicates the degree of connectivity of neurons (Fig. 5g). The total number of lines in the map, as well as the number of nodes with connecting lines, indicate the overall strength of neural networking circuitry throughout this organoid (Fig. 5h), and the number of nodes with connecting lines indicates the quantity of actively networking neurons (Fig. 5i). The increase in these properties over 2 to 6 months of maturation means that neural networking circuitry becomes stronger and more connected, with more neurons communicating interactively. As shown in Fig. 5j, the number of communities increased, while the number of nodes that did not belong to any community decreased. In addition, the average number and the maximum number of nodes included in a single community also increased as this organoid matured (Supplementary Fig. 19). As nodes within one community interact through more robust circuitry than the nodes outside this community, these results indicate that the strength and size of the neural community were enhanced by electrophysiological maturation of this organoid. Long-term monitoring of intra-organoid analysis using our 3D LM MEA can provide the comprehensive information required to fully understand the evolution of the functional activity of brain organoids.

### Magnetic reshaping of 3D LM MEAs for multi-spot detection of intra-organoid signals

Although conventional MEA platforms, including flat surface-type electrodes or 3D electrodes, have been extensively utilized for the electrophysiological analysis of neural tissues, limitations on the number and position of electrodes as well as the inability to adjust their structures remain disadvantages. In our study, by depositing a thin ferromagnetic cobalt (Co) layer (thickness: 120 nm) only on half of the sidewall of each LM pillar before the parylene encapsulation (Supplementary Fig. 20), the 3D LM electrodes could be reshaped by applying external magnetic fields (Fig. 6a). This reshaping capability of the 3D LM electrodes exploited their inherent deformability and softness of EGaIn to prevent these pillars from cracking. To precisely control the tilting of electrodes, we constructed a magnetic tilting system with a 6-axis stage (H-820 6-Axis Hexapod, Physik Instrumente, minimum displacement: 0.5 μm). This system offers minute and uniform controllability over the direction and intensity of the magnetic field through software-based operation (see Supplementary Fig. 21 and Supplementary Movie 3). The magnet (NdFeB magnet, N52 grade, size: 5.5 cm × 5.5 cm × 2.5 cm) was mounted on the other xyz-axis linear stage (M-460P-XYZ, Newport Corporation), which was fixed on the 6-axis stage. Additionally, by integrating the recording interface

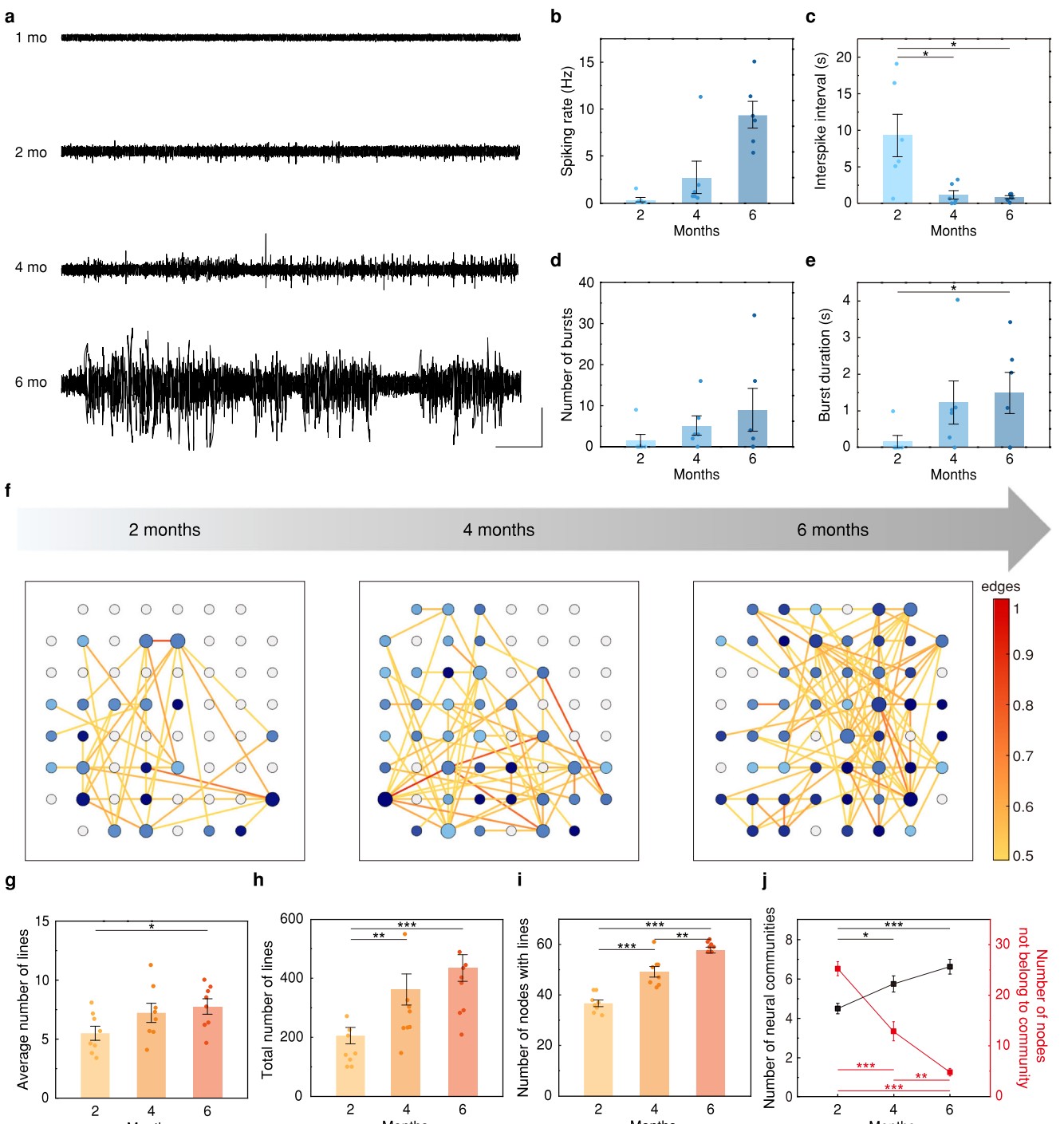

**Fig. 5 | Electrophysiological maturation of cortical organoid. a** SU potential of 1, 2, 4, 6-month-old cortical organoids recorded by the representative electrode located inside each organoid. Scale bars, 100 μV (vertical), 1 s (horizontal). **b–e** Changes in electrophysiological properties during maturation-span of cortical organoids ($n = 6$ independent organoids for each month); (**b**) spiking rate, (**c**) interspike interval ($p = 0.02031$ for 2 and 4 months; $p = 0.01698$ for 2 and 6 months), (**d**) number of bursts, and (**e**) burst durations ($p = 0.03215$ for 2 and 6 months). All data are presented as mean ± s.e.m. Statistical differences were determined with unpaired, one-sided t-test; *$p < 0.05$. **f** Maturation of neural net-working circuitry during maturation-span of cortical organoid. **g–j** Changes in properties related to neural networking circuitry during maturation-span of cortical organoids ($n = 8$ independent organoids per each months); (**g**) average number of connected lines per single node ($p = 0.01140$ for 2 and 6 months), (**h**) total number of connected lines within single organoid ($p = 0.009979$ for 2 and 4 months; $p = 9.715 \times 10^{-5}$ for 2 and 6 months), (**i**) number of nodes with connected lines within single organoid ($p = 8.823 \times 10^{-5}$ for 2 and 4 months; $p = 0.001193$ for 4 and 6 months; $p = 2.692 \times 10^{-10}$ for 2 and 6 months), (**j**) changes in properties related to neural community including number of neural communities ($p = 0.01165$ for 2 and 4 months; $p = 0.0002006$ for 2 and 6 months) and number of nodes not belong to the community within single organoid ($p = 5.704 \times 10^{-5}$ for 2 and 4 months; $p = 0.001194$ for 4 and 6 months; $p = 1.0610 \times 10^{-7}$ for 2 and 6 months). All data are presented as mean ± s.e.m. Statistical differences were determined with unpaired, one-sided t-test; *$p < 0.05$, **$p < 0.01$, and ***$p < 0.001$. SU, single-unit.

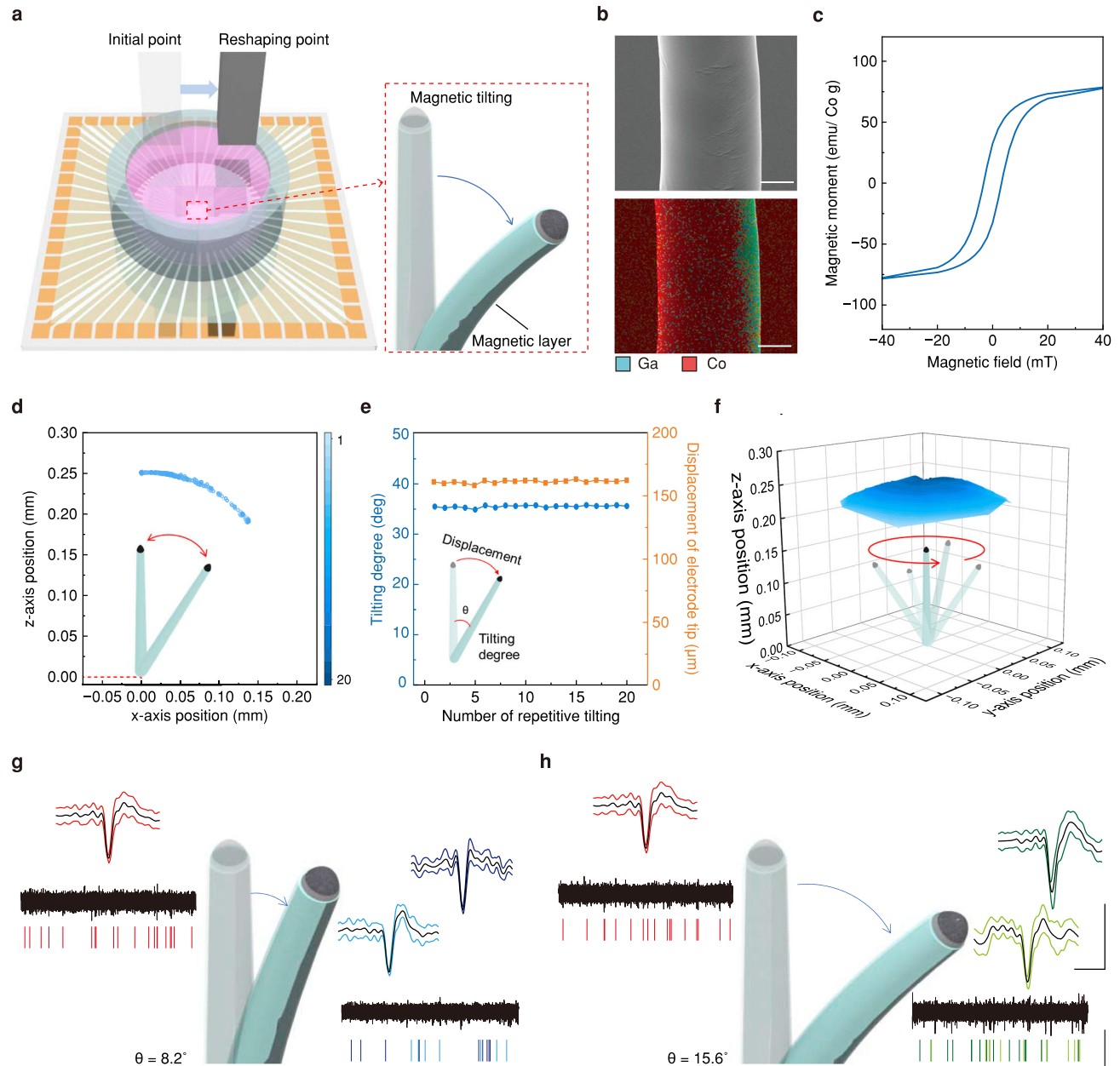

**Fig. 6 | Magnetically reshapable 3D LM MEA. a** Schematic illustration of magnetically reshapable 3D LM MEA. **b** SEM image (top) and EDS analysis (bottom) of partial magnetic layer on 3D LM electrode. Scale bars, 10 μm. This experiment was independently repeated more than ten times with similar results. **c** Magnetization of magnetically reshapable 3D LM electrode. **d** Trajectories of electrode tip during 20 repetitive tilting cycles. The color bar represents the cycle of repetitive tilting. **e** Magnetic tilting degrees (blue) and displacements (yellow) of the electrode tip for the repetitive cycles. **f** Expanded detectable area of the single tiltable electrode in various directions. **g, h** Multi-spot detection of intra-organoid signals by magnetically reshapable 3D LM MEA. SU potential, colored raster plot, and spikeforms when the degree of reshaping (θ) is (**g**) 8.2° and (**h**) 15.6° are presented. Scale bars, 50 μV (vertical), 1 s (horizontal) for single unit potential; 20 μV (vertical), 1 ms (horizontal) for spikeforms. Each color of spikeforms represent individual cluster of spikes. LM, liquid metal; MEA, multi-electrode array; SEM, scanning electron microscopy. EDS, energy-dispersive X-ray spectroscopy. SU, single-unit.

(MZ60, Tucker-Davis Technologies Inc, USA) into the system, we enabled simultaneous recording of intra-organoid signals while tilting the electrodes magnetically. The scanning electron microscopy (SEM) image (top inset in Fig. 6b) and energy dispersive spectroscopy (EDS) analysis (bottom inset) verified this selective deposition of Co only on half of the pillar's sidewall. Figure 6c shows the magnetization of this 3D LM electrode evaluated by a vibrating sample magnetometer (VSM). As a demonstration, with the base of each pillar fixed to the substrate, the position of the electrode's recording site within an organoid shifted according to the direction of the magnetic field,

achieved by tilting the electrode. To investigate the precise controllability of our magnetic tilting system, we theoretically studied the minimum degree of electrode deflection under our system, as shown in Supplementary Note 4. Our system allows the electrode to tilt with a resolution of 0.002°, resulting in a minimum displacement of electrode tip of 0.009 μm. When examining the tilting trajectory of an electrode (height: 260 μm, magnet speed: 5 mm/s) as shown in Supplementary Fig. 22, the tilting degrees of the electrode are linearly increased along the magnet displacement with a slope of 3.34, which represents 0.00334° of tilting per 1 μm of magnet displacement. The

displacements of electrode tip are also changed along the magnet displacement with a slope of 13.99, 0.0140 μm of electrode tip displacement per 1 μm of magnet displacement, which is corresponded to our theoretical study. Owing to the magnetic field induced by the magnetic tilting system, the array of electrodes exhibited reliable tilting with constant tilting degrees (see Supplementary Fig. 23 and Supplementary Movie 4). Considering the size of neuron soma (10–12 μm) as shown in Fig. 3b, c, the electrode tip (electrically open for neural recording) must be shifted at least 24 μm to contact another group of neurons. Since the displacement of the electrode tip by magnetic tilting increases with the electrode height, the electrode's capability to contact other groups of neurons depends on the electrode height under an identical magnetic tilting degree. For example, we calculated the minimum height of the electrode required to achieve a 24 μm displacement with 10° tilting, a feasible tilting degree, using the following Eq. (1);

$$h = \frac{d \times 360°}{2\pi \times \theta} \tag{1}$$

where $h$ is the electrode height, $d$ is the displacement of the electrode tip by tilting, and $\theta$ is the tilting degree. Although an electrode with a height of 138 μm can detect another group of neurons under 10° tilting, we printed 3D LM electrodes with a height of 260 μm for larger displacement to contact numerous neuron groups. To verify the reproducible magnetic tilting of 3D LM electrodes, we conducted 20 cycles of repetitive tilting, sufficient to monitor weekly electrophysiological signals from different neuron groups within a brain organoid over 5 months of maturation. The repetitive tilting was observed through a microscope camera (QImaging Micropublisher 5.0 RTV, Teledyne Photometrics) from the perspective of the xz-plane. The magnet was positioned directly above the 3D LM electrodes (distance between electrodes and the magnet: 2 cm) and tilted the electrodes by shifting the magnet at speed of 5 mm/s until reaching the maximum tilting degree. Then, video analysis software (Tracker, Open Source Physics) was used to track the position of the electrode tip and obtain x, z coordinates for calculating the tilting degree, the angle between the electrodes before and after tilting (Fig. 6d and Supplementary Movie 5). The 260 μm height electrode showed a maximum tilting degree of 35.5 ± 0.05° and a displacement of the electrode tip of 161 μm in a single direction along consistent trajectories over 20 cycles of repetitive tilting (Fig. 6d, e), implying reliable tilting to detect the same group of neurons over multiple cycles of tilting. Furthermore, magnetic tilting in both directions along the identical trajectory in the xz-plane was investigated using the same method. The 3D LM electrode was placed under the magnet with a spacing of 2 cm, and the magnet moved horizontally in both directions at 5 mm/s until reaching the maximum tilting in both directions. The total maximum tilting degree of 3D LM electrodes showed 51.6 ± 0.21°, and the displacement of the electrode tip was 234 μm over 20 cycles of repetitive tilting in both directions (Supplementary Fig. 24 and Supplementary Movie 6). These results indicate that accurate control of the magnetic field by the magnetic tilting system facilitated reproducible tilting of 3D LM electrodes and ensured reliable contact with the same neuron group over multiple tilting cycles. Furthermore, with the system's facile magnet control, the 3D LM electrodes can be tilted in various directions (Supplementary Movie 7). By integrating data from 8 different trajectories into a single graph (Fig. 6f), we inferred the detectable area of a single electrode with magnetic tilting. As a result, the single electrode can cover an expanded area of 6.994×10⁴ μm² within the organoid, indicating that the electrode exhibited about 891 times larger detectable area than that of an electrode without magnetic tilting. Considering the diameter of neurons (10–12 μm) as shown in Fig. 3b and c, this expanded detectable area allows the electrode to reach other groups of neurons.

The detectable areas of the MEAs in previous studies were compared in Supplementary Table 1, validating the superior spatial resolution of our system's recording sites. In addition, SEM images of 3D LM electrodes before and after magnetic tilting showed no vestige of leakage, which might occur by the physical deformation from magnetic tilting, showing the structural stability owing to intrinsic softness and self-healability of the LM (Supplementary Fig. 25).

As the viscosity affects the tilting of electrodes, we studied the relationship between electrode tilting and medium viscosity, as shown in Supplementary Note 5[40]. The viscosity of the medium induces a drag force against the magnetic force exerted on the 3D LM electrode, resulting in a reduction in the tilting speed. To experimentally examine tilting behavior, we used a 0.6 wt% agarose gel instead of brain organoids, as imaging the inside of brain organoids is challenging due to their opacity. Since brain organoids exhibit similar mechanical properties and viscosity compared to the brain, we adopted this transparent 0.6 wt% agarose gel, commonly used as a brain phantom due to its similar mechanical properties, to substitute for brain organoids[41]. When tracking the positions of the electrode tip (height: 250 μm) along the magnet movement at 3 mm/s in air and in the agarose gel (Supplementary Fig. 26 and Supplementary Movie 8), magnetic tilting in this gel (red dots) was slower than in air (blue dots), corresponding to the theoretical study. Although it took more time to overcome the drag force of the viscous medium, the electrode tips reached identical positions (when sufficient time is provided for the electrode to tilt) regardless of the medium's viscosity due to the facile deformability of the 3D LM electrode and sufficient magnetic field to exert the driving force for tilting. This result assures reliable tilting of the electrodes within brain organoids, further demonstrated by the change in the neural recording results.

As the brain organoids are securely immobilized onto the substrate of 3D LM MEAs with the aid of cell-adhesive coating, the brain organoids do not shift under the magnetic tilting of electrodes within them (Supplementary Movie 9). Moreover, we assessed the potential damage caused by the magnetic tilting of 3D LM electrodes within brain organoids. In the magnetic tiling group, the organoids were subjected to an applied magnetic field, causing the electrodes to tilt at an angle of 20° for 3 hours. To evaluate the influence of this tilting on the viability of the brain organoids, we conducted quantitative real-time polymerase chain reaction (qPCR) analysis for the neuronal marker (MAP2), astrocyte marker (S100B), and apoptosis marker (Caspase-3). The results were compared with a control group that was not exposed to the magnetic field. As shown in Supplementary Fig. 27, the expression levels of MAP2, S100B, and CASP3 did not significantly differ between the magnetic tilting group and the control group. The low modulus and high mechanical deformability of LM electrodes contributed to minimizing the risk of damage to the brain organoids. In addition, the 3D visualization of the organoids both without and with magnetic tilting (electrodes at the tilted position) was performed to examine the damage that might be induced due to the 20 cycles of magnetic tilting (magnet speed of 5 mm/s for 4 seconds per each tilting). As shown in Supplementary Fig. 28, the magnetically tiltable electrodes remained intactly and the cytoarchitecture of the organoid was well preserved even after the 20 cycles of tilting. Additionally, a comparison of phosphorylated Tau (p-Tau) expression, a marker of brain damage, confirms that the magnetic tilting of 3D LM electrodes within the organoid does not cause any damage to the inserted organoid while altering the interaction of the electrodes with neural cells, potentially allowing for a broader recording of neural network activity within the organoid.

As shown in Supplementary Fig. 29, the electrode impedance did not change significantly by Co deposition because the electrode tip was preserved without a Co coating. This impedance showed negligible changes under 20 cycles of repetitive magnetic tilting (20.697 ± 0.502 kΩ to 20.781 ± 0.480 kΩ), assuring the stability of electrical performance of magnetically tiltable electrodes

(Supplementary Fig. 30). Also, when we examined the influence of the magnetic field on intra-organoid signal recording, there was a slight increase in the noise level only during magnet movement (maximum noise level: ~ 52 μV), due to changes in the magnetic field. However, the noise level reduced to its initial state again after the movement ceased. Since our electrophysiological recordings were conducted only when the magnet remained stationary, the noise induced during the magnet movement did not affect these recordings significantly (Supplementary Fig. 31). For accurate analysis on the effect of magnetic field on firing rate by excluding any possible deformation of electrodes, we loaded brain organoid on 3D LM MEA with non-magnetic stationary electrodes. Significant changes in spiking rate along the existence of the magnetic field generated by the magnet were not observable, indicating that the magnetic field did not affect on the spontaneous neural activity of the organoids since the strength of the magnet of our system (285 mT on the surface) is not large enough to induce any changes in neural activities while deflecting magnetic tiltable electrodes (Supplementary Fig. 32).

Furthermore, Fig. 6g, h and Supplementary Fig. 33 demonstrate that a single electrode in MEA can collect electrophysiological signals from multiple points within a cortical organoid, effectively densifying the recording sites without needing additional electrodes. Although LFPs and SU potentials did not differ significantly depending on the degree of magnetic tilting, changes in spikeforms were evident due to this tilting. These changes indicated that when the electrode was tilted, it came into contact with a different group of neurons, each exhibiting a unique spikeform. The number of spike clusters was detected as one before tilting, and it became two clusters after tilting (Fig. 6g). By further tilting this electrode, different spikeforms were detected while maintaining the number of clusters (Fig. 6h). To further verify the multi-spot detection by magnetically tilted 3D LM electrodes, we fabricated the non-magnetic stationary electrodes and magnetically tiltable electrodes in a single substrate and recorded neural signals by them from a single organoid (Supplementary Fig. 34). The spiking rates of both non-magnetic electrodes and magnetic electrodes did not show specific tendency in response to the magnetic tilting, indicating that the spiking rates reflect neural activities of brain organoids regardless of the magnetic tilting (Supplementary Fig. 35a). On the other hand, the spike waveform similarity of magnetically tiltable electrodes showed much lower value (0.740) compared to that of non-magnetic stationary electrodes (0.911), showing the detection of different group of neurons by magnetic tilting (Supplementary Fig. 35b). These results indicate that the non-magnetic electrodes are not influenced by the magnetic field gradient generated by magnet movement, while magnetic electrodes shift their recording spots through the magnetic tilting. The change in SU spikes indicated alternation in contact groups of neurons by moving the recording position of the electrode. Therefore, this magnetic reshaping of our 3D LM MEA allows a single electrode to record from multiple spots within the brain organoid, which can be effective in forming high-density recording sites. We also examined intra-organoid signals to verify that the reproducible tiling of the electrode can detect the same group of neurons over 20 tilting cycles. The intra-organoid signals of a 4-month-old organoid loaded onto the 3D LM MEA (electrode height: 260 μm) were recorded during 20 repetitive tilting cycles. The magnet (spacing from MEA: 2 cm) was moved horizontally at a speed of 5 mm/s for 4 seconds, corresponding to an electrode tip displacement of 120 μm, enabling contact with other groups of neurons. To validate the detection of the identical neuron group, similarities in spikeforms were calculated using Pearson's correlation coefficient, as in previous studies[42]. When comparing the spikeforms between the first and the 20th tilting, the signal similarities were 0.923 and 0.973 at the initial position and tilted position, respectively. This indicates that reliable magnetic tilting facilitated reproducible detection of neuron groups over repetitive tilting cycles (Supplementary Fig. 36).

## Discussion

We developed magnetically reshapable 3D LM MEAs for intra-organoid analysis. High-resolution direct printing of the biocompatible LMs enabled the formation of 3D soft electrodes with adjustable configurations of recording sites, leading to the collection of intra-organoid signals across a wide range of 3D coordinates within brain organoids. The softness and fine structure of our electrodes made them suitable for chronic monitoring of neural activity with minimal invasion into organoids. And the flexibility in placement and height of the 3D LM electrodes allowed for customization to address the specific 3D morphology of individual organoids. Our MEAs recorded intra-organoid signals with sufficient spatiotemporal resolutions to extract electrophysiological information such as LFPs, SU potentials, and spikeforms as well as neural circuitry developed in the 3D interior volume of organoids. Furthermore, we demonstrated the magnetic reshaping of 3D LM electrodes for recording from multiple sites within the organoid. This provides a solution that increases the recording density without having to fabricate additional electrodes. Our approach can offer an opportunity to precisely explore the electrophysiological dynamics of the brain and to clarify the origins of neurodegenerative disorders. For instance, integrating our 3D LM MEAs into disease-modeled brain organoids can facilitate the discovery of dysfunctions in the neuropathological circuits of specific diseases. In addition, these MEAs can be utilized to monitor the responses of patient-driven organoids to drugs, further excavating patient-specific drugs. Nonetheless, several potential shortcomings should be addressed to expand the applicability of this technique. Firstly, the printing system uses a single nozzle, making it time-consuming for mass production. Additionally, since the liquid metal is printed under the ambient conditions, this method has structural constraints compared to other printing methods, such as direct laser writing or inkjet printing, which utilize photocurable, thermoset, or rigid materials. While the liquidous properties of EGaIn present these challenges, it also offers irreplaceable advantages including minimized tissue damage and robust magnetic tilting for high-resolution electrophysiological analysis of brain organoids.

Furthermore, we note that challenges still remain, given the need for the development of reliable brain organoids that resemble the full functionality of the human brain. For example, the disparities that exist in neural activity and neuronal density between brain organoids and in-vivo brain tissue impede the acquisition of electrophysiological information from the organoids. Likewise, the results of our intra-organoid signal recording showed abated neural activities in amplitudes and spiking rates in comparison to in-vivo neural signals. In addition, insufficient delivery of oxygen and nutrients to the core of organoids becomes more problematic as their sizes increase, which hinders chronic analysis due to the reduced viability of the organoids. Therefore, in conjunction with the advancement in cultivating robust brain organoids, the use of 3D LM MEAs as an intra-organoid analysis platform provides substantial potential to expand the applicability of brain organoids by offering in-depth insights into their functional activities.

## Methods

### Liquid metal printing

A glass capillary (Sutter Instrument) with an outer diameter of 1.0 mm and an inner diameter of 0.5 mm was pulled with a pipette puller (P-1000, Sutter Instrument) to prepare the nozzle which had an inner diameter of 18 μm. EGaIn (75.5% gallium, 24.5% indium alloy by weight; Changsha Santech Materials Co. Ltd.) was used as an ink for liquid metal printing. All printing steps of EGaIn were monitored using a microscope camera (QImaging Micropublisher 5.0 RTV, Teledyne Photometrics) to control the nozzle position from a substrate using a 6-axis stage (H-820 6-Axis Hexapod, Physik Instrument) during the printing process. In the case of 18 μm-diameter nozzle, using a

compressed dry air, the air pressure of 80 psi was applied to EGaIn to be extruded from the syringe (i.e., an ink reservoir) to the tip of a glass capillary nozzle before starting printing.

## Fabrication of 3D LM MEA
The fabrication steps of the 3D LM MEA are as follows; (1) A slide glass was prepared in size of 4.5 × 4.5 cm. First, interconnect pads were formed by deposition of Cr/ Au and Cr/ Pt through e-beam evaporator (Korea Vacuum Tech Co. Ltd) and photolithography. Then, LM interconnect lines and 3D LM pillars were printed. For passivation, parylene-C were patterned with a shadow mask to open the contact pad for recording system and electrode tip.

## Sidewall passivation of 3D LM pillar
After the printing of 3D LM pillars, 2 μm of parylene-C was deposited on whole surface of the device. Parylene-C on the tip of the pillars was selectively dry-etched by reactive ion etching system with the condition of 200 W, 240 seconds.

## Electrodeposition of platinum nanoparticles
For preparing 25 mL of an electrodepostion solution, 25 mL of iso-propyl alcohol, 5 mg of lead acetate trihydrate (Sigma-Aldrich), and 0.25 g of platinum tetrachloride (Sigma-Aldrich) were mixed at room temperature. This electrodeposition solution was stirred under ultra-sonic vibration for 20 minutes and filtered to remove impurities. The electroplating was performed by ion transfer between the cathode and anode in the Pt electrodeposition solution. A cathode (the 3D LM electrode) and an anode (Pt electrode) were immersed in this elec-trodeposition solution, and each was connected to a source meter (Keithley 2400, Tektronix). The electroplating reaction occurred under an electrical voltage of 2 V and compliance current of 0.1 mA for 120 seconds.

## Attachment of elastomer wells
Elastomer wells were fabricated with polydimethylsiloxane (PDMS). For the macrowell, uncured PDMS was poured in square petri dish with 2 cm-thickness. Once PDMS was cured, 19 mm-diameter hole was created with a punch. For the microwell, micromold was designed with 3D printer (Form3, Formlabs), and uncured PDMS was poured onto the mold. Cured microwell was attached on the 3D LM MEA, and macrowell was attached with the uncured PDMS as an adhesive material.

## Impedance spectroscopy
The impedance measurements of the 3D LM electrodes were con-ducted in a PBS solution (Sigma-Aldrich). All impedance measure-ments were performed over a frequency range of 0.01 to 100 kHz using a multichannel potentiostat (PMC-1000, AMETEK).

## hiPSC maintenance
hiPSC line KYOU-DXR0109B (#ACS-1023; American Type Culture Col-lection, Manassas, VA, USA) was used for cortical organoid generation[43,44]. The use of hiPSCs was approved by the Institutional Review Board (IRB) of Yonsei University (Permit Number: 7001988-202309-BR-1066-02E). hiPSCs were cultured on Matrigel-coated dishes (#354277, Corning) with mTeSR-Plus (#5825, STEMCELL Tech-nologies) and maintained at passages 20 to 45 using ReLeSR (#5872, STEMCELL Technologies). The cells were regularly checked by MycoAlert PLUS Mycoplasma Detection Kit (#LT07-705, Lonza) for mycoplasma contamination and the infection was prevented by Myco-Guard™ Mycoplasma Elimination Reagent (#SMD022, Biomax).

## Generation of cortical organoids
Cortical organoids were generated and cultured with a slightly mod-ified protocol from a previous study[33,45]. hiPSCs were dissociated into single cells with TrypLE Express Enzyme (#12604013, Thermo Fisher Scientific) and transferred to ultra-low attachment 96-well plate (#CLS7007, Corning), with each well containing 10,000 cells in mTeSR-Plus medium supplemented with the ROCK inhibitor Y-27632 (20 μM; #1293823, BioGems). For the first 5 days, the medium was changed every day and supplemented with dorsomorphin (2.5 μM; #P5499, Sigma), SB-431542 (10 μM; #1614, Tocris), and XAV939 (2.5 μM; # X3004, Sigma). On the sixth day in suspension, neural spheroids were transferred to ultra-low attachment 24-well plate (#cls3473, Corning) in neural medium containing neurobasal-A (#0888-022, Thermo Fisher Scientific), 1X B-27 supplement without vitamin A (#12587010, Thermo Fisher Scientific), 1X GlutaMax (#35050061, Thermo Fisher Scientific), 1X Penicillin/Streptomycin (#15140122, Thermo Fisher Scientific) and supplemented with the epidermal growth factor (EGF; 20 ng ml$^{-1}$; #AF100-15, Peprotech) and fibroblast growth factor 2 (FGF2; 20 ng ml$^{-1}$; #100-18B, Peprotech) until day 24. From day 25 to 42, the medium was supplemented with brain-derived neurotrophic factor (BDNF; 20 ng ml$^{-1}$, #450-02, Pepro-tech) and neurotrophin 3 (NT3; 20 ng ml$^{-1}$; #450-03, Peprotech). Medium was changed every other day. From day 43 onward, cortical organoids were maintained in neurobasal-A medium supplemented with 1X B-27 supplement without vitamin A with medium changes every 4–6 days.

## Immunohistochemistry
The cortical organoids were fixed in 10% formalin solution (#HT501640, Sigma) for two hours at room temperature and washed in phosphate-buffered saline (PBS, Biosesang, Seongnam, Korea). Subsequently, they were immersed in 1:1 (v/v) OCT compound (#HCP-0100-00A, CellPath):30% sucrose (#84097, Sigma) dissolved in PBS overnight at 4 °C for cryoprotection. The organoids were embedded in OCT compound and frozen in liquid nitrogen, and sectioned at 10-15-μm-thickness using a cryostat (Leica). Cryosections were washed with PBS to remove excess OCT compound and permeabilized with 0.25% (v/v) Triton X-100 (#X100, Sigma) in PBS for 20 min. Then, the sections were treated with 4% (w/v) bovine serum albumin (#216006980, MP Biomedicals), 2% (v/v) horse serum (#16050130, Thermo Fisher Sci-entific), and 0.02% (v/v) Triton X-100 in PBS for 1 h at room tempera-ture, and incubated with primary antibodies diluted in PBS overnight at 4 °C. The stained samples were then washed with PBS and incubated with Alexa Fluor 488, 555, or 594-conjugated secondary antibodies in PBS (1:200; Thermo Fisher Scientific) for 1 h at room temperature. The nuclei were counterstained with 4′,6-diamidino-2-phenylindole (DAPI; #A2412, TCI America) for 20 min and washed with PBS. The samples were mounted using a fluorescent mounting medium (#H1400, Vector laboratories). Images were acquired with a confocal microscope (LSM 880, Carl Zeiss) and processed with ImageJ (National Institutes of Health). The following primary antibodies were used for immunohis-tochemistry: anti-TUJ1 (mouse, 1:500, #801213, Biolegend), anti-SOX2 (rabbit, 1:200, #AB5603, Millipore), anti-MAP2 (rabbit, 1:100, #4542S, Cell Signaling Technology), anti-GFAP (mouse, 1:200, #MAB3402, Millipore), anti-VGLUT1 (rabbit, 1:500, #135 302, Synaptic Systems), anti-CTIP2 (rat, 1:250, #ab18465, Abcam), and anti-SATB2 (mouse, 1:25, #ab51502, Abcam).

## Assessment of biocompatibility
The viability of cortical organoids with 3D LM electrodes was mea-sured using Live/Dead assay kit (Thermo Fisher Scientific). The staining procedure was conducted following the instructions provided by the manufacturer. The cortical organoids were washed with PBS and stained with Live/Dead solution consisting of 2 μM calcein-AM (to label live cells) and 4 μM ethidium homodimer-1 (to label dead cells) with incubation at 37 °C for 60 minutes, followed by washing with PBS. Fluorescent images of cortical organoids were obtained using a con-focal microscope (LSM 880), and organoid viability was assessed

based on Live/Dead stained images, and analyzed with ImageJ software.

## Real-time quantitative PCR analysis

mRNA was isolated using TaKaRa MiniBEST Universal RNA Extraction Kit (#9767 A; TaKaRa Bio lnc.) and template cDNA was synthesized from the extracted mRNA using TaKaRa PrimeScript II First-strand cDNA synthesis kit (#6110 A; TaKaRa). Subsequently, qPCR analysis was conducted with the cDNA samples, TaqMan Fast Universal PCR MasterMix (#4366073; Thermo), and the following TaqMan gene expression assay kits: PAX6 (Hs00240871_m1), Nestin (Hs04187831_g1), TUBB3 (Hs00801390_s1), MAP2 (Hs00258900_m1), S100B (Hs00902901_m1), and CASP3 (Hs_00234387_m1) using a StepOnePlus Real-Time PCR System (Applied Biosystems). The relative gene expression of each target was calculated by the comparative Ct method and normalized to glyceraldehyde 3-phosphate dehydrogenase (GAPDH; Hs02786624_g1).

## 3D imaging of the cleared cortical organoids

3D imaging of the cleared cortical organoids was performed using a slightly modified protocol from previous studies[46,47]. The organoids were fixed in 10% formalin solution (#HT501640, Sigma) for two hours at room temperature and washed in phosphate-buffered saline (PBS, Biosesang, Seongnam, Korea). The fixed organoids were washed with a washing buffer and treated with a blocking buffer for 8 h at 37 °C. Next, the organoids were incubated with primary antibodies for 67 h at 37 °C. After washing with the washing buffer, the organoids were incubated with secondary antibodies and DAPI solution for 68 h at 37 °C. Following this, the organoids were washed again with the washing buffer and cleared by replacing the Ce3D solution overnight. Images were acquired using a confocal microscope (LSM 700 and 880, Carl Zeiss) and processed with ImageJ (National Institutes of Health).

## Electrophysiological recording

For the data analysis of LFPs and SU potentials, an electrophysiological recording system was used, which consisted of a RZ2 amplifier processor, PZ5 Neurodigitizer, MZ60 MEA interface (Tucker-Davis Technologies Inc, USA), and a computer with Synapse program. A sampling rate of 24,414 Hz and 60 Hz notch were used during recording. Mostly 0.1–300 Hz bandpass filter was used for recording LFPs, and a 300–3,000 Hz bandpass filter was used for recording single-unit spikes. For long-term monitoring of cortical organoids, the cortical organoids of 2, 4, and 6 months were placed on the 3D MEAs respectively, and the neural signals from them were recorded.

## Fabrication of magnetically reshapable 3D LM MEA

Magnetically reshapable 3D LM electrodes were fabricated by deposition of a ferromagnetic layer directly on the electrodes. Before printing 3D LM pillars on the substrate, shadow mask was positioned upon the substrate in order to form magnetic layer only on the 3D LM pillars. The ferromagnetic layer (cobalt) was vacuum deposited only on the half side of LM pillars by tilting the substrate during the deposition. The later processes were proceeded same as the fabrication process of 3D LM MEA.

## VSM analysis

For quantitative analysis, the magnetic hysteresis loops were measured with a vibrating sample magnetometer (VSM, 7404-S, Lake Shore Cryotronics) for all samples. Since Co coated EGaIn was in a liquid state, an appropriate amount ( ~100 mg) was trapped in the cartridge container and fixed to a sample holder to prevent unintended vibration.

## Magnetic tilting system

For precise control of the magnetic field for accurate tilting of electrodes, a magnet (NdFeB magnet, N52 grade, size: 5.5 cm × 5.5 cm ×

2.5 cm) is mounted on an xyz-axis linear stage (M-460P-XYZ, Newport Corporation), which is fixed on the 6-axis stage (H-820 6-Axis Hexapod, Physik Instrumente). The movements of the 6-axis stage (minimum displacement: 500 nm) are controlled by software (LabVIEW, National Instruments). Furthermore, the MEA interface (MZ60, Tucker-Davis Technologies Inc.) is integrated into this system for neural recording of brain organoids.

## Analysis of magnetic tilting of 3D LM electrode

The magnetic tilting of 3D LM electrodes was recorded using a microscope camera (QImaging Micropublisher 5.0 RTV, Teledyne Photometrics) from the perspective of the xz-plane. Subsequently, the tilting was analyzed using a video analysis tool (Tracker, Open Source Physics). The electrode tips were targeted for tracking, providing x, z coordinates. The tilting degree was then calculated using the following Eq. (2);

$$\theta = \tan^{-1}\left(\frac{|x_2 - x_1|}{z_2}\right) \tag{2}$$

where $\theta$ is the tilting degree, $x_1$ is the x-coordinate of the electrode tip before tilting, and $x_2$ and $z_2$ are the x-, z-coordinates of the electrode tip after tilting, respectively.

## Data analysis

**Spike detection and PCA clustering.** All analyses of neural recording data were done using MATLAB R2022b with four open-source toolboxes (Statistics and Machine Learning Toolbox, Signal Processing Toolbox, Bioinformatics Toolbox, and Circular Statistics Toolbox by Philipp Berens). Any timepoint was classified as a spike peak if its signal amplitude was lower than -5 times of the signal standard deviation. A spike waveform was defined as a 4-milliseconds-long slice of the signal centered at a spike peak.

In order to sort the detected spike waveforms that originated from different neurons, we applied dimension reduction and clustering algorithms. We first used principal component analysis (MATLAB function pca) to reduce each high-dimensional spike waveform into vectors residing in a two-dimensional feature space. These vectors were then clustered into pre-specified number of groups, using the k-means algorithm (MATLAB function kmeans). We set the number of groups as 2 or 3.

**Burst detection.** We defined bursts using five pre-defined threshold parameters—maximum interspike interval at start of burst, maximum interspike interval in burst, minimum burst duration, minimum interburst interval, and minimum number of spikes in burst. Any series of spikes that satisfies all of these thresholds was considered as a burst. We set the threshold as 300, 301, 10, 50, 200, respectively.

**2D and 3D neural network map.** We quantified the synchronization level between two electrodes using the SPIKE-distance between the spike trains[35]. For the calculation of SPIKE-distance, we used Python package Pyspike (version 0.7.0) on a Python environment (version 3.9.2)[48]. The SPIKE-distance was normalized to average of 1,000 randomly generated baseline values; each baseline value is the SPIKE-distance between two uniformly randomly generated spike trains. Then, synchronization score was defined as SPIKE-distance values subtracted from 1. This score ranged from 0 to 1, where a higher score indicated a high level of synchrony.

Based on this synchronization score, community structure on the electrodes was examined by graph-theoretic Louvain algorithm which is based on modularity optimization[36,49]. In the analysis, we modelled the electrodes as nodes (e.g., circles in the network map), whose 3D positions represented the position of the actual electrode. We set the

synchronized scores between every pair of active electrodes as edge (e.g., lines in the network map) weights between the corresponding nodes. Edges with synchronized scores less than 0.5 were filtered out. The result of the Louvain algorithm was visualized through color and size, using the Python package plotly (version 5.9.0). Nodes with the same color (i.e., electrodes) represented electrodes belonging to the same neural community. Synchronization scores were color-mapped, ranging from 0 (yellow) to 1 (red). In addition, the node size indicated the number of connected electrodes.

## Statistics and reproducibility

For Fig. 2b–e and Fig. 3a, b, statistical analyzes were conducted using GraphPad Prism 10 (GraphPad, La Jolla, CA, USA). Statistical significance was determined using unpaired, two-sided Student's t-tests and one-way analysis of variance (ANOVA) with Tukey's multiple comparisons tests based on the test requirements. For the other figures, statistical analyzes were conducted using Excel (Microsoft, Washington, USA), and statistical significance was determined using unpaired, one-tailed t tests. All details regarding experimental replications, biological replicates (n), and statistical tests are included in the corresponding figure legends.

## Ethics

This study does not involve experiment involving animals, human participants, or clinical samples.

## Reporting summary

Further information on research design is available in the Nature Portfolio Reporting Summary linked to this article.

## Data availability

All data supporting the findings of this study are available within the article and its supplementary files. Any additional requests for information can be directed to, and will be fulfilled by, the corresponding authors. The data used in this study are available in the Figshare (https://doi.org/10.6084/m9.figshare.24764508)[50]. Source data are provided with this paper for reproducing all Figures in the manuscript and Supplementary Information.

## Code availability

The custom code used for this study is available at Code Ocean (https://doi.org/10.24433/CO.7867348.v1)[51].

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

## Acknowledgements

This work was supported by the Ministry of Science & ICT (MSIT) through the National Research Foundation, STEAM Research Business - Korea Global Cooperative Convergence Research Program (RS-2024-00460364) (J.-U. P) and ERC Program (RS-2024-00406240) (J.-U. P).

## Author contributions

E.K. and E.J. carried out the experiment, analyzed the data, and wrote the manuscript. E.K., Y.-M.H., I.J., Y.W.K., Y.-G.P., and J.-Y.K. were involved in device fabrications and electrophysiological analysis. E.J., J.K., J.L., and S.C. were involved in organoid cultivation and assessment of bio-compatibility. J.-U.P., S.-W.C., and J.-H.L., an oversaw all of the research phases and revised the manuscript. All authors discussed and commented on the manuscript.

## Competing interests

S.-W.C. is a chief technology officer (CTO) of Cellartgen, Inc., Republic of Korea. The remaining authors declare no competing interests.
