## [Transparent Peer Review file · Nature Communications]

Magnetically reshapable 3D multi-electrode arrays of liquid metals for electrophysiological analysis of brain organoids

Corresponding Author: Professor Jang-Ung Park

Version 1:

Reviewer comments:

Reviewer #1

(Remarks to the Author)

Kim et al. designed a 3D MEA for cortical organoids, which can detect neural signals from different locations at different depths of the cortical organoid for a long time. This device is important for detecting signals from 3D neural organoids and is currently a frontier for development in technology. However, there are concerns about the current results before this manuscript is published. Here are some comments.

Major points:

1. Lines 221-223: I am confused by the description of the 3D electrodes being naturally inserted inside the brain organoid. Is it enough to aspirate the culture solution without any other operations? An experimental video of this process should be necessary to help readers understand this design. Additionally, Fig. 3b results from sectioned cortical organoids perpendicular to 3D pillars. The sectioned cortical organoids parallel to 3D pillars would be better because they are more intuitive and effective for demonstrating the effect of inserting different heights of 3D pillars into the brain organoid.
2. Fig. 4c and 4j, Supplementary Figure 7f: Using the synchronization score alone should not be sufficient to identify synchronized activity. Synchrony occurs just as neurons generate action potentials simultaneously on millisecond time scales. Network bursting is defined by alternating periods of high and low activity, a measure of how the spikes from all of the neurons are organized in time. Neither synchrony nor burst is evident in any of the data provided by the authors, e.g., Figs 5a and 4f, and supplementary Figure 7b.
3. Lines 304-313: The cortical organoid can grow a laminar distribution of tissue structures similar to the deep and superficial structures of the cerebral cortex, but this structure is not present in uniform thickness at all locations in the cortical organoid. Moreover, it varies from one cortical organoid to another. For example, this is shown in Fig.2K in Ref.42. So, 3D pillars of the same height are not guaranteed to contact the same layer-specific neurons. Therefore, I am puzzled by the result of Fig. 4i. Why the neural activity detected by 3D pillars of 100 μm height is lower than that detected by 3D pillars of 50 and 20 μm height? Is this result also consistent across different cortical organoids?
4. Fig. 4i and Supplementary Figure 7e: 3D pillars at 100 μm height have the lowest detected neural activity, an interesting result that can easily interest the reader. However, I have two concerns:
(1) Fig. 4i, 4j, Supplementary Figures 7e and 7f are bar charts. Does the height represent the average value? Dot plots are suggested to show the specific values of different channels.
(2) Supplementary Figure 7 provides the neural activity recorded by 3D pillars at different positions, but there is no design of 100 μm pillars, only 50 and 200, so there can not be a direct comparison with Fig. 4i. Combined with my question 3, I am curious if this variation in neural activity is only related to height or to position as well.
5. Does long-term monitoring refer to using the same 3D MEA with the same cortical organoid, with signals recorded at 2, 4, and 6 months? Was the cortical organoid kept on the same 3D MEA continuously for 6 months or only placed on 3D MEA at 2, 4, and 6 months? It is recommended that the authors provide specific details in the Methods section to help the reader better understand the advantages of this 3D MEA. This is critical to demonstrate the stability and effectiveness of the 3D MEA.
6. Page 20: Supplementary Video 3 shows that all the electrodes are tilted together when the magnetic field changes. This

raises a concern about whether the cortical organoid is completely immobile at this time or if it moves together with the tilted electrodes. The authors should provide a video showing this result, which is important for later analyzing the damage produced by the tilted electrodes and the recorded neural activity:

- (1) If the cortical organoid is immobile, the electrode's tilt will damage the organoid. The use of qPCR on page 20 to characterize the damage is not sufficient, and it is recommended to provide the results of fluorescence immunostaining of the sectioned organoid parallel to the electrode to characterize this damage;
- (2) If the cortical organoid moves with the tilted electrode, then the electrode's precision in acquiring signals from different locations needs to be reevaluated.

7. Fig. 6g, 6h and Supplementary Figure 18: Is it possible that the changes in neural activity and spike waveforms are due to the deformation of the organoid caused by tilting the electrode? In Supplementary Figure 18, when comparing the initial and tilted positions, the fluctuations in the tilted position seem to be higher, and the signals after 20 cycles are also higher than those after 1 tilt. This result is more like constant external stimulation, not necessarily recording the signal in different positions. In conjunction with the previous question, the authors need to provide more evidence about the ability of tilted 3D pillars to record signals in different positions.

8. Lines 455-456: Does the tilted electrode cause leakage of EGaIn? How about the stability of the impedance? It is recommended that the authors provide impedance data when the electrode is tilted.

Minor points:

1. Line 65: References 15 and 16 are inappropriate; they are not neural networks in organoids. Suggestions are to cite others or rephrase the sentence.
Line 73: Reference 19 is inappropriate; this paper uses planar MEA to record electrical signals from a complete cerebral organoid, not a sectioned organoid. Suggest citing another or redescribing this sentence.
Line 91: Reference 25 is inappropriate because it is irrelevant to this sentence. Suggest citing another or redescribing this sentence.
Lines 92-94: Can a reference or evidence be provided to support this sentence by demonstrating that this stretchable and fine design can reduce the inflammatory response?
2. Lines 71-73: There is also the calcium imaging method, suggesting improving the description.
3. Lines 170-173: Fig. 1g shows no less than 4x, suggesting a rephrase of this result.
4. Fig. 3d has a poor-quality image.
5. Supplementary Figure 6b only shows two points; is it counting only two channels? It is suggested that data from more channels be provided to prove that the spiking rate significantly increases under the stimulation of KCl.
6. The author designed the device with 60 channels, but why does Fig. 4c analyze the synchrony between 64 channels?
7. Page 31: There is a red font.

(Remarks on code availability)

Reviewer #2

(Remarks to the Author)

In this work, the authors report 3D-printed needle-like multi-electrode arrays to record single units within the brain organoid. There are some elements of novelty, especially with the ability for magnetic reshapability and dynamic recordings. A significant limitation of the paper is that the study on magnetic reshaping is limited (Fig. 6), and a more in-depth study on the significance and capabilities of this reshapability would strengthen the paper. The following issues should be addressed.

1. If the magnetic movement shifts the entire set of needles by a small amount in one direction, couldn't the same result be obtained by pushing the organoids slightly? Please justify. The novelty of using magnetically driven deflection should be the ability to move the needles in other complex configurations.
2. Some studies talk about the effect of magnetic fields on changes in neural activity and ion channels, so it might be good to show a study (besides the change in the noise level that the authors presented) where the firing before and after the magnet is introduced.
3. It would be good to analyze the power spectral density of the LFP data further to see any peculiar effects.
4. Based on the fabrication steps provided, could the researchers fabricate magnetic needles first and then nonmagnetic on the same chip to simultaneously record using both needles to see how the magnetic field affects unit detection? If that experiment results in no changes in nonmagnetic electrodes and detection of different units before and after deflection in magnetic electrodes, that would be more conclusive.

5. Authors should discuss the potential shortcomings of the presented method. For instance, direct laser writing methods can make complex shapes, albeit at a slower throughput. Also, the current designs can only do inter-organoid recording from the bottom half of the organoid. Would it be possible to print curved needles to collect data from the top half of the organoids (similar to the interconnects the authors presented in their earlier studies)?

6. Organoid placement strategy – Media is aspirated after placing the organoid on top of the needle array to “settle down” the organoid. Authors claim that the electrodes are “naturally” inserted into the organoid during this process. It is unclear how much of the media is aspirated and whether capillary forces drive the insertion or the gravity. If it is the capillary, does it affect the needles themselves (elastocapillary effect) since they are soft?

7. While the increase in firing rate following the KCl introduction could indicate that the activity is from organoids, having a negative control by using neurotoxins like TTX could further confirm that the captured data is not noise. Especially since the reported noise level (13 microvolts) is not particularly low.

In conclusion, the novelties of the presented work are as follows:

1. It can create needle-like electrodes of different heights with contact points only at the tip using a scalable direct writing method, providing some new insight into volumetric electrophysiology.
2. Through modifications, these electrodes can be deflected using an external magnetic field. The authors show that the deflected electrodes detect different units.
3. It is possible to fabricate high-throughput MEA platforms capable of getting electrophysiology data from many organoids (nine).

However, the main article does not focus enough on these points, especially on the new capabilities enabled by magnetic reshaping, and more in-depth studies are needed to validate the utility of this approach.

However, the main article does not focus enough on these points, especially on the new capabilities enabled by magnetic reshaping, and more in-depth studies are needed to validate the utility of this dynamic and changeable recording approach.

(Remarks on code availability)

Reviewer #3

(Remarks to the Author)

(Remarks on code availability)

Version 2:

Reviewer comments:

Reviewer #1

(Remarks to the Author)

In this revised manuscript, the authors have responded well to the reviewer's requests in rebuttal, there are some points that the authors would better consider as follows;

1. Supplementary Figure 18: What is the 3D morphology of the organoids after 7 weeks? Should the changes in morphology before and after 7 weeks be presented? Given that the authors coated the device with PDL-Laminin, which promotes neural cell adhesion and migration, is the 3D morphology of the organoids still intact after 7 weeks?
2. Lines 180 and 504: The unit of the impedance data presented in the text should be $k\Omega$ rather than Ω .

(Remarks on code availability)

Reviewer #2

(Remarks to the Author)

The authors have reasonably addressed issues and the paper can be accepted.

(Remarks on code availability)

[Response to Reviewer#1]

We appreciate your thoughtful review of this manuscript with valuable comments, and we welcome the opportunity to address and clarify the issues. Our response to the points raised in the comment is as follows:

Comment 1: “Lines 221-223: I am confused by the description of the 3D electrodes being naturally inserted inside the brain organoid. Is it enough to aspirate the culture solution without any other operations? An experimental video of this process should be necessary to help readers understand this design. Additionally, Fig. 3b results from sectioned cortical organoids perpendicular to 3D pillars. The sectioned cortical organoids parallel to 3D pillars would be better because they are more intuitive and effective for demonstrating the effect of inserting different heights of 3D pillars into the brain organoid.”

Response to Comment 1:

For better understanding of the organoid insertion process into the 3D liquid metal (LM) multi-electrode array (MEA), we recorded the procedure as shown in newly added Supplementary Video 2. Initially, the organoid is placed into the 3D LM MEA device, which contains culture media. At this stage, the 3D LM electrodes are densely distributed, and the organoid is light enough to rest gently on top of the electrode array without crushing them. Then, the media is gradually aspirated using a micropipette (aspirated media volume: 1000 μ l), thereby eliminating the buoyancy from the media. This leaves gravity as the dominant acting on the organoid. As the organoid settles due to gravity, the organoid adheres to the bottom of the device, with the electrodes naturally inserting into the organoid during this process. Once the media is replenished, the organoid remains securely attached to the device, even when shaken. After subsequent culturing, the organoid becomes firmly attached to both the electrodes and the bottom of the device, which is coated with poly-L-lysine (PLL) and laminin. We have added the video (Supplementary Video 2) showing the insertion process and the robust attachment of brain organoids onto the device.

Furthermore, to effectively demonstrate the insertion of 3D LM electrodes into the brain organoids, we performed optical clearing of the organoid followed by whole-mount 3D imaging (Supplementary Figure 6). The organoids were integrated onto the 3D LM MEA (electrode height: 150 μ m) and stained with TUJ1 and DAPI for further investigation of internal architecture of the organoid, which could potentially be deformed by the inserted electrodes.

[**Supplementary Figure 6 (newly added).** Whole-mount 3D imaging of optically cleared organoids without (left) and with (right) inserted 3D LM electrodes. The organoids were stained with TUJ1 and DAPI for investigation of internal architecture of organoids.]

By analyzing the entire region where the electrodes were inserted, we confirmed that the 3D LM electrodes were inserted into the organoid straight, maintaining their structure. The organoid also preserved its characteristic rosette structure, which is a specific structure of brain organoids, indicating the absence of structural damage to the organoid. This suggests that the 3D LM electrodes can be stably inserted into the organoid without significantly compromising their structural integrity, allowing for reliable monitoring of neural network within the organoid through the inserted 3D LM electrodes. We have supplemented these results as Supplementary Figure 6 and explanation in the revised manuscript to reinforce the reliability of our insertion process.

Revised manuscript (page 11):

Fig. 3a and **Supplementary Video 2** illustrates the integration of this cortical organoid to our 3D LM MEA.

Revised manuscript (pages 11):

Furthermore, we performed optical clearing of the organoid followed by whole-mount 3D imaging (Supplementary Fig. 6), showing that the 3D LM electrodes were inserted into the organoid straight without causing any structural damage on brain organoids, allowing for reliable monitoring of neural network within the organoid.

Revised manuscript (page 33):

3D imaging of the cleared cortical organoids

3D imaging of the cleared cortical organoids was performed using a slightly modified protocol from previous studies^{48,49}. The organoids were fixed in 10% formalin solution (#HT501640, Sigma) for two hours at room temperature and washed in phosphate-buffered saline (PBS, Biosesang, Seongnam, Korea). The fixed organoids were washed with a washing buffer and treated with a blocking buffer for 8 h at 37 °C. Next, the organoids were incubated with primary antibodies for 67 h at 37 °C. After washing with the washing buffer, the organoids were incubated with secondary antibodies and DAPI solution for 68 h at 37 °C. Following this, the organoids were washed again with the washing buffer and cleared by replacing the Ce3D solution overnight. Images were acquired using a confocal microscope (LSM 700 and 880, Carl Zeiss) and processed with ImageJ (National Institutes of Health).

Revised Supplementary Information (page 8):

Supplementary Figure 6. Whole-mount 3D imaging of optically cleared organoids without (left) and with (right) inserted 3D LM electrodes. The organoids were stained with TUJ1 and DAPI for investigation of internal architecture of organoids.

Revised Supplementary Information (page 41):

Supplementary Video 2. Organoid insertion process.

Comment 2: “Fig. 4c and 4j, Supplementary Figure 7f: Using the synchronization score alone should not be sufficient to identify synchronized activity. Synchrony occurs just as neurons generate action potentials simultaneously on millisecond time scales. Network bursting is defined by alternating periods of high and low activity, a measure of how the spikes from all of the neurons are organized in time. Neither synchrony nor burst is evident in any of the data provided by the authors, e.g., Figs 5a and 4f, and supplementary Figure 7b.”

Response to Comment 2:

As the reviewer mentioned, synchronized activities are defined by simultaneous activation of neurons, which are feasible when the neurons are functionally connected. To evaluate the functional connectivity of neural tissues, several studies have utilized synchronization scores [*Nat. Commun.*, 12, 492, 2021]. In our manuscript, we measure the synchronization scores to quantify the neural connectivity developed within the brain organoids [*J. Neurophysiol.*, 109, 1457, 2013]. However, as the reviewer pointed out, presenting raw traces of neural signals is essential to validate that the method we employed to examine the neural connectivity reliably reflects the neural activities of brain organoids. Thus, we have added representative traces of single-unit potentials from the brain organoid which shows the synchronized activities (Supplementary Figure 10).

[**Supplementary Figure 10 (newly added)**. Representative synchronized activities of the brain organoid recorded by 3D LM MEA. Red arrows indicate synchronized activities. a, Single unit potentials from 8 individual electrodes. Scale bars, 100 μ V (vertical), 0.5 s (horizontal). b, Raster plots from 8 individual electrodes. Scale bar, 0.5 s.]

In our manuscript, bursts are characterized when the single-unit potential shows rapid spiking [Stem cells, 34, 1040, 2016]. The specific criteria for burst, including maximum interspike interval, minimum burst duration, minimum interburst interval and minimum number of spikes in burst, are detailed in Methods section. Similar to the synchronization score, presenting raw traces of single-unit potentials showing bursting activities can further reinforce our results. Thus, we also have added the raw traces of single-unit potentials from brain organoids that show bursting activities (Supplementary Figure 12).

[**Supplementary Figure 12 (newly added)**. Representative bursts of the brain organoid recorded by 3D LM MEA. Blue arrows indicate bursts of neural activities from the organoid. Scale bars, 50 μ V (vertical), 0.5 s (horizontal).]

Revised manuscript (pages 13-14):

The representative synchronized activities of the brain organoid are presented in Supplementary Fig. 10.

Revised manuscript (pages 15):

In addition, the neural activity of this organoid along the electrode height was analyzed by the parameters of spiking rate and burst number (Fig. 4i, j, and Supplementary Fig. 12).

Revised Supplementary Information (page 12):

Supplementary Figure 10. Representative synchronized activities of the brain organoid recorded by 3D LM MEA. Red arrows indicate synchronized activities. **a**, Single unit potentials from 8 individual electrodes. Scale bars, 100 μ V (vertical), 0.5 s (horizontal). **b**, Raster plots from 8 individual electrodes. Scale bar, 0.5 s.

Revised Supplementary Information (page 14):

Supplementary Figure 12. Representative bursts of the brain organoid recorded by 3D LM MEA. Blue arrows indicate bursts of neural activities from the organoid. Scale bars, 50 μ V (vertical), 0.5 s (horizontal).

Comment 3: “Lines 304-313: The cortical organoid can grow a laminar distribution of tissue structures similar to the deep and superficial structures of the cerebral cortex, but this structure is not present in uniform thickness at all locations in the cortical organoid. Moreover, it varies from one cortical organoid to another. For example, this is shown in Fig.2K in Ref.42. So, 3D pillars of the same height are not guaranteed to contact the same layer-specific neurons. Therefore, I am puzzled by the result of Fig. 4i. Why the neural activity detected by 3D pillars of 100 μm height is lower than that detected by 3D pillars of 50 and 20 μm height? Is this result also consistent across different cortical organoids?”

Response to Comment 3:

As the reviewer mentioned, the cellular layers within a single organoid are not uniformly developed, reflecting the randomness of the cytoarchitecture of organoids. In this context, the 3D LM MEAs composed of various heights of electrodes at desired positions, which are presented in Fig. 1d and Supplementary Figure 1, demonstrate the capability of positioning recording sites at various 3D coordinates within the organoids, which are not intended to present the targetability of our electrodes to specific cellular layers of the organoid. Owing to our distinct virtue of customizing electrode configurations, the recording sites of the 3D LM MEA are outspread throughout the organoid. Consequently, we could monitor the neural activities across the entire organoid (newly added Supplementary Figure 11), which shows varied spiking rates depending on the 3-dimensionally distributed spots.

[**Supplementary Figure 11 (newly added).** Intra-organoid signals recorded by 3D LM MEA with 3-level of height variance. The spiking rates were observed at various 3D coordinates across the organoid.]

The data shown in Fig. 4i represent the average spiking rates recorded by 5 adjacent electrodes of the same height. We inferred that these 5 adjacent electrodes in the same height detected neural signals from a localized point within the organoid, and the variations in spiking rates across different electrode heights reflect the differences in the degrees of the neural activities at the distinct points within the 3D organoid. These results show the capability of 3D LM MEAs to observe the intra-organoid signals from different regions within the 3D organoid.

Given the variability in cytoarchitecture between brain organoids, the results differ to other organoids when monitoring neural activities with 3D LM MEAs, which have the same electrode configuration (3 levels of heights) as employed for obtaining data of Fig. 4i. The recording results are presented in newly added Supplementary Figures 13, 14. For instance, in Fig. 4i, the spiking rate of the 100 μm -height electrodes is lower than that of the 50 μm and 200 μm height-electrodes, whereas in Supplementary Figure 13, the spiking rates increase with electrode height. Additionally, in another organoid (Supplementary Figure 14), the 100 μm -height electrodes show the lowest spiking rate, while the 200 μm -height electrodes show the highest spiking rates. These results indicate that the facile controllability of the electrode height of our 3D printing system facilitates the positioning of recording sites across the organoids and monitoring of neural activities at various 3D coordinates within the organoids, which differs between individual organoids.

[**Supplementary Figure 13 (newly added)**. Intra-organoid signals recorded by 3D LM MEA with 3-level of height variance from another organoid. Each color represents the height of the electrodes; Orange, 50 μm , green, 100 μm , and blue, 200 μm . **a**, Representative single-unit potentials depending on the heights of the electrodes. Scale bars, 20 μV (vertical), 0.5 s (horizontal). **b**, Representative spikeforms depending on the heights of the electrodes. Scale bars, 20 μV (vertical), 0.5 ms (horizontal). **c,d**, Changes in neural activities depending on the heights of the electrodes ($n = 5$ electrodes per each height); (**c**) Spiking rate, (**d**) Number of bursts. All data are presented as mean \pm s.e.m.]

[**Supplementary Figure 14 (newly added)**. Intra-organoid signals recorded by 3D LM MEA with 3-level of height variance from the other organoid. Each color represents the height of the electrodes; Orange, 50 μm , green, 100 μm , and blue, 200 μm . **a**, Representative single-unit potentials depending on the heights of the electrodes. Scale bars, 20 μV (vertical), 0.5 s (horizontal). **b**, Representative spikeforms depending on

the heights of the electrodes. Scale bars, 20 μV (vertical), 0.5 ms (horizontal). **c**, Changes in spiking rates depending on the heights of the electrodes ($n = 5$ electrodes per each height). All data are presented as mean \pm s.e.m.]

Similarly, using the 3D LM MEAs with 2 different heights of electrodes, we monitored neural activities from other organoids as shown in newly added Supplementary Figures 16, 17. Each organoid shows distinct neural activities regardless of electrode heights, showing that the 3D LM MEA captures the intra-organoid signals arisen from the randomly developed cytoarchitecture which is inconsistent across different organoids. We have included these data as Supplementary Figures 13, 14, 16, and 17 in the revised manuscript.

[**Supplementary Figure 16 (newly added)**. Intra-organoid signals recorded by 3D LM MEA with 2-level of height variance from another organoid. Each color represents the height of the electrodes; Orange, 50 μm , and blue, 200 μm . **a**, Representative single-unit potentials depending on the heights of the electrodes. Scale bars, 50 μV (vertical), 0.5 s (horizontal). **b**, Representative spikeforms depending on the heights of the electrodes. Scale bars, 50 μV (vertical), 1 ms (horizontal). **c,d**, Changes in neural activities depending on the heights of the electrodes ($n = 5$ electrodes per each height); (**c**) Spiking rate, (**d**) Number of bursts. All data are presented as mean \pm s.e.m.]

[**Supplementary Figure 17 (newly added)**. Intra-organoid signals recorded by 3D LM MEA with 2-level of height variance from the other organoid. Each color represents the height of the electrodes; Orange, 50 μm , and blue, 200 μm . **a**, Representative single-unit potentials depending on the heights of the electrodes. Scale bars, 50 μV (vertical), 0.5 s (horizontal). **b**, Representative spikeforms depending on the heights of the electrodes. Scale bars, 50 μV (vertical), 1 ms (horizontal). **c**, Changes in spiking rates depending on the heights of the electrodes ($n = 5$ electrodes per each height). All data are presented as mean \pm s.e.m.]

Consequently, these results indicate that the flexibility of designing electrode configuration facilitates monitoring of neural signals from the various 3D coordinates across the brain organoids, leading to the comprehensive understanding of functionality of the brain organoids.

Revised manuscript (page 14):

For example, we printed 3D LM pillars with 3 different heights to record SU potentials and spikes at various coordinates within a single organoid (Fig. 4e and Supplementary Fig. 11).

Revised manuscript (page 15):

The neural activities from other organoids monitored by identical electrode configuration are shown in Supplementary Figs. 13, 14. Given the variability in cytoarchitecture between brain organoids, the results differ to other organoids. For instance, in Fig. 4i, the spiking rate of the 100 μm -height electrodes is lower than that of the 50 μm and 200 μm height-electrodes, whereas in Supplementary Fig. 13, the spiking rates increase with electrode height. A similar tendency was also observed in the analysis of the intra-organoid signals obtained using another 3D LM MEA with 2 different heights of electrodes (Supplementary Figs. 715-17). The variations in intra-organoid signals corresponding to the electrode heights are consistent with layer-specific expression of discriminative neural markers (Fig. 2f, g) and findings from a previous report³⁵³⁸, which demonstrated distinct electrical properties for each sub-neuronal type constituting the cortical layer. These results indicate that the ability of our 3D LM MEA to detect various 3D coordinates within the organoid facilitates capturing variations in neural activities throughout the organoid, which is inconsistent across different organoids due to the randomly developed cytoarchitectures. Also, these analyses along the electrode height as well as 3D neural networking map provide insight into the neural circuitry between neurons across cortical layers, a capability that has not been achievable with previous approaches using surface-type MEAs.

Revised Supplementary Information (page 13):

Supplementary Figure 11. Intra-organoid signals recorded by 3D LM MEA with 3-level of height variance. The spiking rates were observed at various 3D coordinates across the organoid.

Revised Supplementary Information (page 15):

Supplementary Figure 13. Intra-organoid signals recorded by 3D LM MEA with 3-level of height variance from another organoid. Each color represents the height of the electrodes; Orange, 50 μm, green, 100 μm, and blue, 200 μm. **a**, Representative single-unit potentials depending on the heights of the electrodes. Scale bars, 20 μV (vertical), 0.5 s (horizontal). **b**, Representative spikeforms depending on the heights of the electrodes. Scale bars, 20 μV (vertical), 0.5 ms (horizontal). **c,d**, Changes in neural activities depending on the heights of the electrodes ($n = 5$ electrodes per each height); (**c**) Spiking rate, (**d**) Number of bursts. All data are presented as mean \pm s.e.m.

Revised Supplementary Information (page 16):

Supplementary Figure 14. Intra-organoid signals recorded by 3D LM MEA with 3-level of height variance from the other organoid. Each color represents the height of the electrodes; Orange, 50 μm, green, 100 μm, and blue, 200 μm. **a**, Representative single-unit potentials depending on the heights of the electrodes. Scale bars, 20 μV (vertical), 0.5 s (horizontal). **b**, Representative spikeforms depending on the heights of the electrodes. Scale bars, 20 μV (vertical), 0.5 ms (horizontal). **c**, Changes in

spiking rates depending on the heights of the electrodes ($n = 5$ electrodes per each height). All data are presented as mean \pm s.e.m.

Revised Supplementary Information (page 19):

Supplementary Figure 16. Intra-organoid signals recorded by 3D LM MEA with 2-level of height variance from another organoid. Each color represents the height of the electrodes; Orange, 50 μm , and blue, 200 μm . **a**, Representative single-unit potentials depending on the heights of the electrodes. Scale bars, 50 μV (vertical), 0.5 s (horizontal). **b**, Representative spikeforms depending on the heights of the electrodes. Scale bars, 50 μV (vertical), 1 ms (horizontal). **c,d**, Changes in neural activities depending on the heights of the electrodes ($n = 5$ electrodes per each height); **(c)** Spiking rate, **(d)** Number of bursts. All data are presented as mean \pm s.e.m.

Revised Supplementary Information (page 20):

Supplementary Figure 17. Intra-organoid signals recorded by 3D LM MEA with 2-level of height variance from the other organoid. Each color represents the height of the electrodes; Orange, 50 μm , and blue, 200 μm . **a**, Representative single-unit potentials depending on the heights of the electrodes. Scale bars, 50 μV (vertical), 0.5 s (horizontal). **b**, Representative spikeforms depending on the heights of the electrodes. Scale bars, 50 μV (vertical), 1 ms (horizontal). **c**, Changes in spiking rates depending on the heights of the electrodes ($n = 5$ electrodes per each height). All data are presented as mean \pm s.e.m.

Comment 4-1: “Fig. 4i and Supplementary Figure 7e: 3D pillars at 100 μm height have the lowest detected neural activity, an interesting result that can easily interest the reader. However, I have two concerns:

(1) Fig. 4i, 4j, Supplementary Figures 7e and 7f are bar charts. Does the height represent the average value? Dot plots are suggested to show the specific values of different channels.”

Response to Comment 4-1:

The heights of the bars in Fig. 4i, 4j and Supplementary Figures 7e, 7f (original manuscript, which is revised as Supplementary Figure 15e, 15f in the revised manuscript) represent the average values from 5 electrodes. As the reviewer suggested, we have modified the bar charts in these figures to dot plots.

[Fig. 4i, j (revised). Changes in neural activity depending on the heights of the electrodes ($n = 5$ electrodes per each height); (i) Spiking rate, (j) Number of bursts.]

[Supplementary Figure 15 e,f (revised). Changes in neural activity depending on the heights of the electrodes ($n = 5$ electrodes per each height); (e) Spiking rate, (f) Number of bursts. All data are presented as mean \pm s.e.m.]

Revised manuscript (pages 51-53):

Fig 4. | Intra-organoid signals from 4-month-old cortical organoids. **a,b**, Representative intra-organoid signals of 4-month-old cortical organoid integrated on 3D LM MEA; **(a)** Single unit (SU) potential. Scale bars, 50 μ V (vertical), 5 s (horizontal). **(b)** Spikeforms. Scale bars, 50 μ V (vertical), 2 ms (horizontal). **c**, Color mapping of synchronization score between the electrodes located inside the 4-month-old cortical organoid. **d**, Neural networking map based on synchronization score and neural community of 4-month-old cortical organoid integrated on 3D LM MEA. **e**, Configuration of 3D LM electrodes with 3-level of heights. **f,g**, Representative intra-organoid signals depending on the heights of the electrodes. Each color represents the height of the electrodes; Orange, 50 μ m, green, 100 μ m, blue, 200 μ m. **(f)** SU potentials. Scale bars, 100 μ V (vertical), 0.5 s (horizontal). **(g)** Spikeforms. Scale bars, 100 μ V (vertical), 0.5 ms (horizontal). **h**, 3D neural networking map corresponding to the 3-level height variance of 3D LM MEA. **i,j**, Changes in neural activity depending on the heights of the electrodes ($n = 5$ electrodes per each height); **(i)** Spiking rate, **(j)** Number of bursts. All data are presented as mean \pm s.e.m.

Revised Supplementary Information (pages 17-18):

Supplementary Figure 715. Analysis on intra-organoid signals recorded by 3D LM MEA with 2-level of height variance. **a**, Configuration of 3D LM electrodes with 2-level of heights. **b,c**, Representative intra-organoid signals depending on the heights of the electrodes. Each color represents the height of the electrodes; Orange, 50 μm , blue, 200 μm . **(b)** SU potentials. Scale bars, 100 μV (vertical), 0.5 s (horizontal). **(c)** Spikeforms. Scale bars, 100 μV (vertical), 0.5 ms (horizontal). **d**, 3D neural networking map corresponding to the 2-level height variance of 3D LM MEA. **e,f**, Changes in neural activity depending on the heights of the electrodes ($n = 5$ electrodes per each height); **(e)** Spiking rate, **(f)** Number of bursts. All data are presented as mean \pm s.e.m.

Comment 4-2: “(2) Supplementary Figure 7 provides the neural activity recorded by 3D pillars at different positions, but there is no design of 100 μm pillars, only 50 and 200, so there can not be a direct comparison with Fig. 4i. Combined with my question 3, I am curious if this variation in neural activity is only related to height or to position as well.”

Response to Comment 4-2:

We present Supplementary Figure 15 (Supplementary Figure 7 of the original manuscript) to show the flexibility of our 3D printing system for developing various configurations of electrodes within the brain organoids. The ability to customize the electrode arrays enables the overall monitoring of neural activities as well as the neural networking across the organoid. In this context, we designed the electrode array composed of 3 heights of electrodes in Fig. 4 and 2 heights in Supplementary Figure 15 to show the feasibility of the customization of electrodes, enabling the monitoring of neural activities in 3D coordinates within the organoid as shown in newly added Supplementary Figure 11. Since the cytoarchitecture differs from one organoid to the other as mentioned in reviewer’s comment 3, comparing the recording results of Fig. 4i and Supplementary Figure 15 is inappropriate.

As we have described in Response to Comment 3, the neural activities detected by 3D LM MEAs with same electrode configuration (2 levels or 3 levels of heights) vary to other organoids as shown in Fig. 4, Supplementary Figures 13-17. These results indicate that the variations in neural activities originate from the ability of our 3D LM MEA to detect various coordinates within the organoid by designing the position and the height of the electrodes. We have included the recording results from different organoids in Supplementary Figures 13, 14, 16, 17 and the descriptions to present the ability to capture variations in neural activities throughout the organoid by customizing the electrode configurations.

Revised manuscript (page 15):

The neural activities from other organoids monitored by identical electrode configuration are shown in Supplementary Figs. 13, 14. Given the variability in cytoarchitecture between brain organoids, the results differ to other organoids. For instance, in Fig. 4i, the spiking rate of the 100 μm -height

electrodes is lower than that of the 50 μm and 200 μm height-electrodes, whereas in Supplementary Fig. 13, the spiking rates increase with electrode height. A similar tendency was also observed in the analysis of the intra-organoid signals obtained using another 3D LM MEA with 2 different heights of electrodes (Supplementary Figs. 715-17). The variations in intra-organoid signals corresponding to the electrode heights are consistent with layer-specific expression of discriminative neural markers (Fig. 2f, g) and findings from a previous report³⁵³⁸, which demonstrated distinct electrical properties for each sub-neuronal type constituting the cortical layer. These results indicate that the ability of our 3D LM MEA to detect various 3D coordinates within the organoid facilitates capturing variations in neural activities throughout the organoid, which is inconsistent across different organoids due to the randomly developed cytoarchitectures. Also, these analyses along the electrode height as well as 3D neural networking map provide insight into the neural circuitry between neurons across cortical layers, a capability that has not been achievable with previous approaches using surface-type MEAs.

Revised manuscript (pages 51-53):

Fig 4. | Intra-organoid signals from 4-month-old cortical organoids. **a,b**, Representative intra-organoid signals of 4-month-old cortical organoid integrated on 3D LM MEA; **(a)** Single unit (SU) potential. Scale bars, 50 μ V (vertical), 5 s (horizontal). **(b)** Spikeforms. Scale bars, 50 μ V (vertical), 2 ms (horizontal). **c**, Color mapping of synchronization score between the electrodes located inside the 4-month-old cortical organoid. **d**, Neural networking map based on synchronization score and neural community of 4-month-old cortical organoid integrated on 3D LM MEA. **e**, Configuration of 3D LM electrodes with 3-level of heights. **f,g**, Representative intra-organoid signals depending on the heights of the electrodes. Each color represents the height of the electrodes; Orange, 50 μ m, green, 100 μ m, blue, 200 μ m. **(f)** SU potentials. Scale bars, 100 μ V (vertical), 0.5 s (horizontal). **(g)** Spikeforms. Scale bars, 100 μ V (vertical), 0.5 ms (horizontal). **h**, 3D neural networking map corresponding to the 3-level height variance of 3D LM MEA. **i,j**, Changes in neural activity depending on the heights of the electrodes ($n = 5$ electrodes per each height); **(i)** Spiking rate, **(j)** Number of bursts. All data are presented as mean \pm s.e.m.

Revised Supplementary Information (page 15):

Supplementary Figure 13. Intra-organoid signals recorded by 3D LM MEA with 3-level of height variance from another organoid. Each color represents the height of the electrodes; Orange, 50 μ m, green, 100 μ m, and blue, 200 μ m. **a**, Representative single-unit potentials depending on the heights of the electrodes. Scale bars, 20 μ V (vertical), 0.5 s (horizontal). **b**, Representative spikeforms depending on the heights of the electrodes. Scale bars, 20 μ V (vertical), 0.5 ms (horizontal). **c,d**, Changes in neural activities depending on the heights of the electrodes ($n = 5$ electrodes per each height); **(c)** Spiking rate, **(d)** Number of bursts. All data are presented as mean \pm s.e.m.

Revised Supplementary Information (page 16):

Supplementary Figure 14. Intra-organoid signals recorded by 3D LM MEA with 3-level of height variance from the other organoid. Each color represents the height of the electrodes; Orange, 50 μm, green, 100 μm, and blue, 200 μm. **a**, Representative single-unit potentials depending on the heights of the electrodes. Scale bars, 20 μV (vertical), 0.5 s (horizontal). **b**, Representative spikeforms depending on the heights of the electrodes. Scale bars, 20 μV (vertical), 0.5 ms (horizontal). **c**, Changes in spiking rates depending on the heights of the electrodes ($n = 5$ electrodes per each height). All data are presented as mean \pm s.e.m.

Revised Supplementary Information (page 19):

Supplementary Figure 16. Intra-organoid signals recorded by 3D LM MEA with 2-level of height variance from another organoid. Each color represents the height of the electrodes; Orange, 50 μm, and blue, 200 μm. **a**, Representative single-unit potentials depending on the heights of the electrodes. Scale bars, 50 μV (vertical), 0.5 s (horizontal). **b**, Representative spikeforms depending on the heights of the electrodes. Scale bars, 50 μV (vertical), 1 ms (horizontal). **c,d**, Changes in neural activities

depending on the heights of the electrodes ($n = 5$ electrodes per each height); (c) Spiking rate, (d) Number of bursts. All data are presented as mean \pm s.e.m.

Revised Supplementary Information (page 20):

Supplementary Figure 17. Intra-organoid signals recorded by 3D LM MEA with 2-level of height variance from the other organoid. Each color represents the height of the electrodes; Orange, 50 μm , and blue, 200 μm . **a**, Representative single-unit potentials depending on the heights of the electrodes. Scale bars, 50 μV (vertical), 0.5 s (horizontal). **b**, Representative spikeforms depending on the heights of the electrodes. Scale bars, 50 μV (vertical), 1 ms (horizontal). **c**, Changes in spiking rates depending on the heights of the electrodes ($n = 5$ electrodes per each height). All data are presented as mean \pm s.e.m.

Comment 5: “Does long-term monitoring refer to using the same 3D MEA with the same cortical organoid, with signals recorded at 2, 4, and 6 months? Was the cortical organoid kept on the same 3D MEA continuously for 6 months or only placed on 3D MEA at 2, 4, and 6 months? It is recommended that the authors provide specific details in the Methods section to help the reader better understand the advantages of this 3D MEA. This is critical to demonstrate the stability and effectiveness of the 3D MEA.”

Response to Comment 5:

For long-term monitoring of cortical organoids, the cortical organoids of 2, 4, and 6 months were placed on the 3D MEAs respectively, and the neural signals from them were recorded. Then, we collectively analyzed electrophysiological properties to investigate the development of the functionality of cortical organoids, suggesting the availability of our system to detect the minute changes of neural signals during their maturation. As the reviewer suggested, we have added the specific details describing the process of recording neural signals from the cortical organoids at different maturation stages in Methods section. Furthermore, to show the long-term usability of our 3D LM MEA, we examined the stability of electrical performance of our system. We adopted the accelerated aging-test to effectively imitate the biological environment in shortened period. Based on the previous studies [*Appl. Sci.*, 15, 2311, 2022], placing the electrodes at 87 $^{\circ}\text{C}$ for 12 days is identical to 12 months in the incubator (37 $^{\circ}\text{C}$) where the brain organoids integrated on 3D LM MEAs are stored. As shown in newly added Supplementary Figure 4, the impedance of 3D LM electrodes at 1 kHz were negligibly changed from $18.005 \pm 0.083 \Omega$ to $18.774 \pm 0.145 \Omega$ during 12 days at 87 $^{\circ}\text{C}$, showing the feasibility of the 3D MEA to monitor continuous advancement of electrophysiological properties of brain organoids.

[**Supplementary Figure 4 (newly added)**. Accelerated aging-test of 3D LM electrodes to investigate the functional stability for long-term neural monitoring of brain organoids. All data are presented as mean \pm s.e.m.]

During the revision of this manuscript, we have recorded spontaneous neural activities from a single organoid. The organoid at day 62 of culture was inserted onto the 3D LM MEA and have been monitored for 7 weeks after the insertion, which has been inevitably halted due to the limited period of the revision. The representative single-unit potentials and spikeforms of each week are presented in Supplementary Figure 18a, b, respectively. As shown in Supplementary Figure 18c, d, the amplitudes of spikes and spiking rates increase as the organoid grows, implying that the 3D LM MEA concretely captures the electrophysiological maturation of the organoid. These results show the feasibility for the long-term usage of this platform as a neural monitoring system, which is ensured by the softness and stable performance of 3D LM electrodes. We have added these results in the revised manuscript to show the long-term usability of our 3D LM MEA as electrophysiological monitoring system of brain organoids.

[**Supplementary Figure 18 (newly added)**. Long-term continuous monitoring of neural signals from a single brain organoid for 7 weeks. **a**, Representative single-unit potentials of each week. Scale bars, 20 μ V (vertical), 0.5 s (horizontal). **b**, Representative spikeforms of each week. Scale bars, 20 μ V (vertical), 1 ms (horizontal). **c,d**, Changes in neural signals for 7 weeks; **(c)** Amplitude of spikes, **(d)** Spiking rates. All data are presented as mean \pm s.e.m.]

Revised manuscript (page 9):

Furthermore, when we investigated the stability of the electrical performance of 3D LM electrodes by accelerated aging test, the impedance showed negligible changes for 12 days at 87 $^{\circ}$ C ($18.005 \pm 0.083 \Omega$ to $18.774 \pm 0.145 \Omega$), which corresponds to 12 months in the incubator (37 $^{\circ}$ C) (Supplementary Fig. 4).

Revised manuscript (page 16):

These features of cellular maturation lead to electrophysiological maturation, as presented in Fig. 5 and Supplementary Fig. 18.

Revised manuscript (page 33):

For long-term monitoring of cortical organoids, the cortical organoids of 2, 4, and 6 months were placed on the 3D MEAs respectively, and the neural signals from them were recorded.

Revised Supplementary Information (page 6):

Supplementary Figure 4. Accelerated aging-test of 3D LM electrodes to investigate the functional stability for long-term neural monitoring of brain organoids. All data are presented as mean \pm s.e.m.

Revised Supplementary Information (page 21):

Supplementary Figure 18. Long-term continuous monitoring of neural signals from a single brain organoid for 7 weeks. **a**, Representative single-unit potentials of each week. Scale bars, 20 μ V (vertical), 0.5 s (horizontal). **b**, Representative spikeforms of each week. Scale bars, 20 μ V (vertical), 1 ms (horizontal). **c,d**, Changes in neural signals for 7 weeks; **(c)** Amplitude of spikes, **(d)** Spiking rates. All data are presented as mean \pm s.e.m.

Comment 6: “Page 20: Supplementary Video 3 shows that all the electrodes are tilted together when the magnetic field changes. This raises a concern about whether the cortical organoid is completely immobile at this time or if it moves together with the tilted electrodes. The authors should provide a video showing this result, which is important for later analyzing the damage produced by the tilted electrodes and the recorded neural activity:

(1) If the cortical organoid is immobile, the electrode's tilt will damage the organoid. The use of qPCR on page 20 to characterize the damage is not sufficient, and it is recommended to provide the results of fluorescence immunostaining of the sectioned organoid parallel to the electrode to characterize this damage;

(2) If the cortical organoid moves with the tilted electrode, then the electrode's precision in acquiring signals from different locations needs to be reevaluated.”

Response to Comment 6:

For immobilization of brain organoids onto the 3D LM MEAs, we coat 3D LM MEA with cell-adhesive layers, which is composed of poly-D-lysine and laminin, before integrating brain organoids into the electrodes. This coating guarantees the fixation of brain organoids on the 3D LM MEA, preventing the movement of brain organoids which can be occurred due to the external causes including the flow of medium or magnetic tilting of the electrodes. The detailed procedure of organoid insertion process is presented in newly added Supplementary Video 2. To confirm the immobilization of organoids, we filmed the video showing the brain organoids integrated on the 3D LM MEA during magnetic tilting of electrodes. For the magnetic tilting, we moved the magnet (distance from the MEA: 2 cm) at the speed of 2 mm/s for 4 seconds. As shown in newly added Supplementary Video 9, the brain organoid was securely immobilized on the 3D LM MEA during the magnetic tilting of the electrodes, enabling the electrodes to be shifted and ensuring acquiring signals from different spots within brain organoids.

As the reviewer pointed out, to compare the damage of the organoid inserted with the magnetically tiltable 3D LM electrodes (height: 150 μ m, thickness of cobalt layer: 120 nm), we performed 3D visualization of the organoids both without and with magnetic tilting. The electrodes were tilted repetitively for 20 cycles by moving the magnet (distance from the MEA: 2 cm) at speed of 5 mm/s for 4 seconds each cycle of tilting. The organoids with magnetic tilting were fixed with the 3D LM electrodes at the tilted position and stained with TUJ1, phosphorylated Tau (p-Tau), and DAPI. Whole-mount 3D staining analysis revealed that the 3D LM electrodes remained intactly inserted into the organoid in both conditions, and the overall architecture of the organoid was well preserved. Additionally, comparison of p-Tau expression, a marker of brain damage, confirmed that magnetic tilting did not significantly damage the organoid. This suggests that magnetic tilting of 3D LM electrodes does not cause significant damage to the organoid inserted with electrodes while altering the interaction of the electrodes with neural cells, potentially allowing for a broader recording of neural network activity within the organoid. We have included data showing the immobilization of brain organoids during the magnetic tilting in Supplementary Video 2 and 9, as well as the figure (Supplementary Figure 28) illustrating the absence of damage by the magnetic tilting.

[**Supplementary Figure 28 (newly added)**. Whole-mount 3D imaging of optically cleared organoids without (left) and with (right) magnetic tilting of 3D LM electrodes. For the organoid with magnetic tilting, the organoid was fixed with the 3D LM electrodes at the tilted position. The organoids were stained with

TUJ1, phosphorylated Tau (p-Tau), and DAPI for investigation of internal architecture and damage of organoids.]

Revised manuscript (page 22):

As the brain organoids are securely immobilized onto the substrate of 3D LM MEAs with the aid of cell-adhesive coating, the brain organoids do not shift under the magnetic tilting of electrodes within them (Supplementary Video 9).

Revised manuscript (pages 22-23):

In addition, the 3D visualization of the organoids both without and with magnetic tilting (electrodes at the tilted position) was performed to examine the damage that might be induced due to the 20 cycles of magnetic tilting (magnet speed of 5 mm/s for 4 seconds per each tilting). As shown in Supplementary Fig. 28, the magnetically tiltable electrodes remained intactly and the cytoarchitecture of the organoid was well preserved even after the 20 cycles of tilting. Additionally, a comparison of phosphorylated Tau (p-Tau) expression, a marker of brain damage, confirms that the magnetic tilting of 3D LM electrodes within the organoid does not cause any damage to the inserted organoid while altering the interaction of the electrodes with neural cells, potentially allowing for a broader recording of neural network activity within the organoid.

Revised Supplementary Information (page 31):

Supplementary Figure 28. Whole-mount 3D imaging of optically cleared organoids without (left) and with (right) magnetic tilting of 3D LM electrodes. For the organoid with magnetic tilting, the organoid was fixed with the 3D LM electrodes at the tilted position. The organoids were stained with TUJ1, phosphorylated Tau (p-Tau), and DAPI for investigation of internal architecture and damage of organoids.

Revised Supplementary Information (page 41):

Supplementary Video 1. Direct printing of LM interconnects and 3D pillars.

Supplementary Video 2. Organoid insertion process.

Supplementary Video 23. Magnetic tilting system for simultaneous neural recording.

Supplementary Video 34. Magnetic tilting of 3D LM MEA.

Supplementary Video 45. Repetitive magnetic tilting of 3D LM electrode.

Supplementary Video 56. Magnetic tilting in both directions along the identical trajectory in xz-plane.

Supplementary Video 67. Magnetic tilting of electrodes from multi-perspectives.

Supplementary Video 78. Magnetic tilting of a 3D LM electrode in air and in a 0.6 wt% agarose gel.

Supplementary Video 9. Immobilization of brain organoid under magnetic tilting of electrode.

Comment 7: “Fig. 6g, 6h and Supplementary Figure 18: Is it possible that the changes in neural activity and spike waveforms are due to the deformation of the organoid caused by tilting the electrode? In Supplementary Figure 18, when comparing the initial and tilted positions, the fluctuations in the tilted position seem to be higher, and the signals after 20 cycles are also higher than those after 1 tilt. This result is more like constant external stimulation, not necessarily recording the signal in different positions. In conjunction with the previous question, the authors need to provide more evidence about the ability of tilted 3D pillars to record signals in different positions.”

Response to Comment 7:

We theoretically studied and presented the feasibility of our magnetically tiltable 3D LM electrodes to shift the recording spot within the organoid as shown in Supplementary Note 5 and Supplementary Figure 26 in the revised manuscript. Additionally, as we have responded to comment 6, magnetic tilting of the electrodes does not induce any deformation on the brain organoids, which were confirmed by the immobilization of the organoid on the 3D LM MEA, qPCR analysis, and the immunostaining images (newly added Supplementary Video 9, Supplementary Figures 27, 28). Therefore, we have proposed that the magnetic tilting of 3D LM electrodes enables multi-spot detection within the organoid. To further reinforce the feasibility of multi-spot detection by magnetically tilted 3D LM electrodes, we fabricated the non-magnetic stationary electrodes and magnetically tiltable electrodes in a single substrate and recorded neural signals by them from a single organoid. Based on the single-unit potentials detected by both non-magnetic and magnetic electrodes, we analyzed the spiking rates, spikeforms, and spike waveform similarity. The representative single-unit potentials and spikeforms are presented in newly added Supplementary Figure 34.

[**Supplementary Figure 34 (newly added)**. Neural activities (single-unit potentials, raster plot, and spikeforms) detected by both non-magnetic stationary electrodes and magnetic electrodes from a single organoid. Each row presents representative single unit potential (top), raster plot (middle), and spikeforms (bottom) from a single electrode. The left column shows the neural activities before the magnetic tilting and the right column shows them after the magnetic tilting. Scale bars for single-unit potentials, 50 μ V (vertical), 0.5 s (horizontal). Scale bars for spikeforms, 20 μ V (vertical), 1 ms (horizontal).]

As shown in Supplementary Figure 35a, the spiking rates of both non-magnetic electrodes and magnetic electrodes did not show specific tendency in response to the magnetic tilting, indicating that the spiking rates reflect neural activities of brain organoids regardless of the magnetic tilting. For a more detailed investigation of spikeform changes, we calculated the spike waveform similarities between spikes from the same electrode before and after the magnetic tilting (Supplementary Figure 35b). Negligible changes were observed in the spikeforms from non-magnetic electrodes before and after the magnetic tilting as further verified by spike waveform similarity of 0.911, suggesting that the non-magnetic electrodes consistently detected the same group of neurons, unaffected by magnetic tilting. In contrast, the spikeforms detected by the magnetic electrodes differ after the magnetic tilting, showing the spike waveform similarity of 0.740, which indicates detection of different group of neurons. These results present that the magnetic electrodes shift their recording spots through the magnetic tilting, while the non-magnetic electrodes are not influenced by the magnetic field gradient generated by magnet movement. By simultaneously monitoring the neural signal changes from both non-magnetic and magnetic electrodes within the same organoid along with magnet movement, we reliably confirmed that the magnetically tiltable electrodes can detect different groups of neurons by shifting the recording site, not by external stimulations, without introducing additional changes related to the presence of magnetic field. We have supplemented these data as Supplementary Figures 34, 35 and descriptions in revised manuscript to further verify the multi-spot detection by our magnetically tiltable 3D LM electrodes.

[**Supplementary Figure 35 (newly added)**. Comparison of neural activities between non-magnetic electrodes and magnetically tiltable electrodes. **a**, Changes in spiking rates before and after the magnetic tilting. **b**, Changes in spike waveform similarities between before and after the magnetic tilting for each electrode ($p = 0.02353$). The similarities of spike waveforms of magnetically tiltable electrodes show lower value than that of non-magnetic stationary electrodes. This result indicates that the magnetically tiltable electrodes effectively shift the recording spots within the organoid to detect the other group of neurons, whereas the non-magnetic stationary electrodes consistently detect the same group of neurons, unaffected by the magnetic tilting. All data are presented as mean \pm s.e.m. and statistical differences were determined with unpaired, one-sided t -test; $*p < 0.05$.]

Revised manuscript (page 24):

To further verify the multi-spot detection by magnetically tilted 3D LM electrodes, we fabricated the non-magnetic stationary electrodes and magnetically tiltable electrodes in a single substrate and recorded neural signals by them from a single organoid (Supplementary Fig. 34). The spiking rates of both non-magnetic electrodes and magnetic electrodes did not show specific tendency in response to the magnetic tilting, indicating that the spiking rates reflect neural activities of brain organoids regardless of the magnetic tilting (Supplementary Fig. 35a). On the other hand, the spike waveform similarity of magnetically tiltable electrodes showed much lower value (0.740) compared to that of non-magnetic stationary electrodes (0.911) (Supplementary Fig. 35b), showing the detection of

different group of neurons by magnetic tilting. These results indicate that the non-magnetic electrodes are not influenced by the magnetic field gradient generated by magnet movement, while magnetic electrodes shift their recording spots through the magnetic tilting.

Revised Supplementary Information (pages 37-38):

Supplementary Figure 34. Neural activities (single-unit potentials, raster plot, and spikeforms) detected by both non-magnetic stationary electrodes and magnetic electrodes from a single organoid. Each row presents representative single unit potential (top), raster plot (middle), and spikeforms (bottom) from a single electrode. The left column shows the neural activities before the magnetic tilting and the right column shows them after the magnetic tilting. Scale bars for single-unit potentials, 50 μ V (vertical), 0.5 s (horizontal). Scale bars for spikeforms, 20 μ V (vertical), 1 ms (horizontal).

Revised Supplementary Information (page 39):

Supplementary Figure 35. Comparison of neural activities between non-magnetic electrodes and magnetically tiltable electrodes. **a**, Changes in spiking rates before and after the magnetic tilting. **b**, Changes in spike waveform similarities between before and after the magnetic tilting for each electrode ($p = 0.02353$). The similarities of spike waveforms of magnetically tiltable electrodes show lower value than that of non-magnetic stationary electrodes. This result indicates that the magnetically tiltable electrodes effectively shift the recording spots within the organoid to detect the other group of neurons, whereas the non-magnetic stationary electrodes consistently detect the same group of neurons, unaffected by the magnetic tilting. All data are presented as mean \pm s.e.m. and statistical differences were determined with unpaired, one-sided t -test; $*p < 0.05$.

Comment 8: “Lines 455-456: Does the tilted electrode cause leakage of EGaln? How about the stability of the impedance? It is recommended that the authors provide impedance data when the electrode is tilted.”

Response to Comment 8:

As the reviewer suggested, we investigated the stability of EGaln electrodes against the magnetic tilting. To examine the possible leakage of EGaln by magnetic tilting, we observed the electrodes (diameter: 20 μm , height: 250 μm , thickness of cobalt: 120 nm) which had undergone 20 cycles of magnetic tilting. As shown in newly added Supplementary Figure 25, the wrinkles were formed on the surface of the electrodes, which might be caused by repetitive tilting. However, there was no vestige of EGaln leakage. This structural stability against the physical deformation would be derived from the intrinsic softness and self-healability of EGaln.

[**Supplementary Figure 25 (newly added).** Scanning electron microscopy (SEM) images magnetically tiltable electrode; **(a)** Before magnetic tilting, **(b)** After 20 cycles of repetitive magnetic tilting. Scale bars, 10 μm .]

Furthermore, to confirm the functional stability, we monitored the impedance of the electrodes at the initial position and at the tilted positions during the 20 cycles of the magnetic tilting. As shown in newly added

Supplementary Figure 30, the electrodes maintained their impedances during the 20 cycles of the magnetic tilting, which shows the maximum variance of 1.083% among 5 electrodes. When comparing impedance before and after the 20 cycles of the magnetic tilting, the impedance changed from $20.697 \pm 0.502 \Omega$ to $20.781 \pm 0.480 \Omega$. The electrodes show impedance change of 0.406% between before and after repetitive magnetic tilting, assuring the stability of electrical performance of magnetically tiltable electrodes. We have included these data in the revised manuscript to show the structural and functional stability of magnetically tiltable electrodes.

[**Supplementary Figure 30 (newly added).** Functional stability of 3D LM electrodes over 20 cycles of repetitive magnetic tilting ($n = 5$ electrodes). The impedance of the electrodes shows negligible change under repetitive magnetic tilting. All data are presented as mean \pm s.e.m. and statistical differences were determined with unpaired, one-sided t -test; n.s. indicates not significant.]

Revised manuscript (page 21):

In addition, SEM images of 3D LM electrodes before and after magnetic tilting showed no vestige of leakage, which might occur by the physical deformation from magnetic tilting, showing the structural stability owing to intrinsic softness and self-healability of the LM (Supplementary Fig. 25).

Revised manuscript (page 23):

This impedance showed negligible changes under 20 cycles of repetitive magnetic tilting ($20.697 \pm 0.502 \Omega$ to $20.781 \pm 0.480 \Omega$), assuring the stability of electrical performance of magnetically tiltable electrodes (Supplementary Fig. 30).

Revised Supplementary Information (page 28):

Supplementary Figure 25. Scanning electron microscopy (SEM) images magnetically tiltable electrode; (a) Before magnetic tilting, (b) After 20 cycles of repetitive magnetic tilting. Scale bars, 10 μ m.

Revised Supplementary Information (page 33):

Supplementary Figure 30. Functional stability of 3D LM electrodes over 20 cycles of repetitive magnetic tilting ($n = 5$ electrodes). The impedance of the electrodes shows negligible change under repetitive magnetic tilting. All data are presented as mean \pm s.e.m. and statistical differences were determined with unpaired, one-sided t -test; n.s. indicates not significant.

Minor comment 1-1: “Line 65: References 15 and 16 are inappropriate; they are not neural networks in organoids. Suggestions are to cite others or rephrase the sentence.”

Response to Minor comment 1-1:

We have rephrased the sentence for clarity.

Revised manuscript (page 4):

Specifically, neurons in the brain organoid communicate with each other, and they develop strong neural connections, which allows them to build intricate neural networking circuitry **as in other neural tissues**^{15,16}.

Minor comment 1-2: “Line 73: Reference 19 is inappropriate; this paper uses planar MEA to record electrical signals from a complete cerebral organoid, not a sectioned organoid. Suggest citing another or redescribing this sentence.”

Response to Minor comment 1-2:

We have removed Reference 19 to avoid the confusion for readers.

Revised manuscript (page 4):

Additionally, ~~the~~ conventional electrophysiological analysis of organoids at neural network levels has utilized planar surface-type electrodes without protruding 3D shapes, requiring the sectioning of the organoids in order to provide close contact with the intra-organoidal cells^{18,19}.

Revised manuscript (page 38):

~~19. Fair, S. R. et al. Electrophysiological Maturation of Cerebral Organoids Correlates with Dynamic Morphological and Cellular Development. *Stem Cell Rep.* **15**, 855–868 (2020).~~

Minor comment 1-3: “Line 91: Reference 25 is inappropriate because it is irrelevant to this sentence. Suggest citing another or redescribing this sentence.”

Response to Minor comment 1-3:

In reference 25 (in the original manuscript, which has been revised as reference 24), the authors examined the mechanical injury in a brain organoid. As the insertion of the rigid electrodes can induce the mechanical damage to brain organoids, we have redescribed the sentence in accordance with the content of the reference.

Revised manuscript (page 5):

The mechanical damage due to the insertion of the rigid electrodes activates ~~The activation of the~~ glial cells ~~around the rigid electrodes~~ and disrupts the chronic monitoring of the neural activity of the organoids^{24,25,27,28}.

Minor comment 1-4: “Lines 92-94: Can a reference or evidence be provided to support this sentence by demonstrating that this stretchable and fine design can reduce the inflammatory response.?”

Response to Minor comment 1:

We apologize for inducing misunderstanding by our statement. The authors of reference 26 (in the original manuscript, which has revised as reference 25) showed that the mesh electronics do not trigger any inflammatory responses compared to the control organoid without mesh electronics (Figure S15 in the reference paper). As the solid evidence that shows the mesh electronics reduces inflammatory responses compared to other electronics is absent, we have revised the related sentence for clarity.

Revised manuscript (page 5):

Although these meshes use solid-state materials, their stretchable design and fineness are beneficial in that they ~~reduce~~~~minimize~~ inflammatory responses, which allows the reliable detection of the neural signals.

Minor comment 2: “Lines 71-73: There is also the calcium imaging method, suggesting improving the description.”

Response to Minor comment 2:

We have introduced the calcium imaging method as an optical technique to monitor neuronal activities.

Revised manuscript (page 4):

Calcium imaging, which employs fluorescent calcium indicators to monitor neuronal activity of the neurons, has also been utilized in brain organoid studies. However, this optical technique is limited by shallow penetration depth caused by the high scattering properties of brain organoids and low temporal resolution, which is unsuitable for examining functional connectivity within the brain organoids. Additionally, ~~the~~ conventional electrophysiological analysis of organoids at neural network levels has utilized planar surface-type electrodes without protruding 3D shapes, requiring the sectioning of the organoids in order to provide close contact with the intra-organoidal cells^{18,49}.

Minor comment 3: “Lines 170-173: Fig. 1g shows no less than 4x, suggesting a rephrase of this result.”

Response to Minor comment 3:

We sincerely apologize for the confusion as we mistakenly included wrong graphs. We have revised Fig. 1g and Supplementary Figure 29 with accurate data.

[**Fig. 1g (revised)**. Comparison of impedances between pristine 3D LM electrode and Pt nanocluster coated LM electrode ($n = 9$ electrodes for each group; $p = 2.27 \times 10^{-13}$). The error bars represent the s.e.m. Statistical differences were determined with unpaired, one-sided t-test; *** $p < 0.001$.]

[**Supplementary Figure 29 (revised)**. Comparison of impedance between magnetically tiltable 3D LM electrodes and pristine 3D LM electrodes ($n = 9$ electrodes for each group). The error bars represent the standard error of the mean (s.e.m.). Statistical differences were determined with unpaired, one-sided t-test; n.s. indicates not significant.]

Revised manuscript (pages 43-45):

Fig 1. | 3D multi-electrode array of liquid metals as an electrophysiological monitoring platform.
a, Schematic illustration of intra-organoid signal recording by 3D LM MEA. **b**, Schematic illustration of printing system and printing process. **c**, Various heights of 3D LM pillars by control printing velocity. Inset presents SEM image, which was taken by 30° tilting of the substrate, presenting 3D LM pillars with various heights. Scale bar, 100 μm. The error bars represent the s.e.m. **d**, Optical images of LM interconnects and 3D LM pillars with height-variance. Scale bars, 100 μm. **e**, Schematic illustrations of 3D LM MEA, and single 3D LM electrode. **f**, SEM image of platinum nanoclusters coated on tip of the electrode (The substrate is 30° tilted when the image was taken). Scale bar, 1 μm. **g**, Comparison of impedances between pristine 3D LM electrode and Pt nanocluster coated LM electrode ($n = 9$ electrodes for each group; $p = 2.27 \times 10^{-13}$). The error bars represent the s.e.m. Statistical differences were determined with unpaired, one-sided *t*-test; *** $p < 0.001$.

Revised Supplementary Information (page 32):

Supplementary Figure 1529. Comparison of impedance between magnetically tiltable 3D LM electrodes and pristine 3D LM electrodes ($n = 9$ electrodes for each group). The error bars represent the standard error of the mean (s.e.m.). Statistical differences were determined with unpaired, one-sided t-test; n.s. indicates not significant.

Minor comment 4: “Fig. 3d has a poor-quality image.”

Response to Minor comment 4:

Since Fig. 3d is quantitative data of Fig. 3c, we assumed your comment referred to Fig. 3c. Therefore, we have replaced Fig. 3c with a better image.

[**Fig. 3c (revised).** Biocompatibility of the 3D LM MEA on cortical organoids; live/dead fluorescence images. Scale bars, 50 μm .]

Revised manuscript (pages 48-50):

Fig 3. | Integration of cortical organoids to 3D LM MEA. **a**, (top) Schematic illustration and (bottom) representative bright-field image of cortical organoids integrated with 3D LM MEA. Scale bar, 500 μm . **b**, Immunofluorescence images of the sectioned cortical organoids with 3D LM electrodes for (top) Nestin and (bottom) TUJ1. Dashed circles represent the circumference of the 3D LM electrode. Scale bars, 50 μm . **c-d**, Biocompatibility of the 3D LM MEA on cortical organoids; (**c**) live/dead fluorescence images. **Scale bars, 50 μm .** and (**d**) viability of cortical organoids with or without the 3D LM electrodes at days 1, 7, and 14 of culture after the insertion. Dashed circles represent the circumference of the 3D LM electrode ($n = 4$). **Scale bars, 50 μm .** **e**, The qPCR analysis of cortical organoids with or without 3D LM MEA at days 7 and 14 of culture after the insertion ($n = 4$). All data are presented as mean \pm s.e.m. and statistical differences were determined with unpaired, two-sided t -test.

Minor comment 5: “Supplementary Figure 6b only shows two points; is it counting only two channels? It is suggested that data from more channels be provided to prove that the spiking rate significantly increases under the stimulation of KCl.”

Response to Minor comment 5:

As the reviewer suggested, we have supplemented additional data from more channels showing significant increase of neural activity by stimulation of 50 mM KCl (Supplementary Figure 8).

[**Supplementary Figure 8 (revised)**. Electrophysiological responses of cortical organoids to KCl treatment. **a**, Single unit potentials and raster plots recorded by 5 individual electrodes. Scale bars, 50 μ V (vertical), 1 s (horizontal). **b**, Comparison in spiking rates of 5 individual electrodes before and after 50 mM KCl treatment ($p = 0.002258$). All data are presented as mean \pm s.e.m. and statistical differences were determined with unpaired, one-sided t -test; ** $p < 0.01$.]

Revised manuscript (page 13):

The 10 mM of KCl was added to the culture medium for intra-organoid recordings, and electrophysiological responses were analyzed as shown in Supplementary Fig. 68a. The spiking rate significantly increased by this KCl treatment, indicating that the recorded intra-organoid signals originated from the activity of neurons (Supplementary Fig. 68b).

Revised Supplementary Information (page 10):

Supplementary Figure 68. Electrophysiological responses of cortical organoids to KCl treatment. **a**, Single unit potentials and raster plots recorded by 35 individual electrodes (top) and spikeforms before and after the treatment (bottom). Scale bars, 50 μ V (vertical), 1 s (horizontal). **b**, Comparison in spiking rates of 2 individual 3D LM MEA 5 individual electrodes before and after 50 mM KCl treatment ($p = 0.002258$). All data are presented as mean \pm s.e.m. and statistical differences were determined with unpaired, one-sided t -test; $**p < 0.01$.

Minor comment 6: "The author designed the device with 60 channels, but why does Fig. 4c analyze the synchrony between 64 channels?"

Response to Minor comment 6:

The recording interface (MZ 60, Tucker-Davis Technologies Inc, USA) is composed of 64 channels including 4 of reference channels. For clarity, we have revised the graph in Fig. 4c.

[Fig. 4c (revised). Color mapping of synchronization score between the electrodes located inside the 4-month-old cortical organoid.]

Revised manuscript (pages 51-53):

Fig 4. | Intra-organoid signals from 4-month-old cortical organoids. **a,b**, Representative intra-organoid signals of 4-month-old cortical organoid integrated on 3D LM MEA; **(a)** Single unit (SU) potential. Scale bars, 50 μ V (vertical), 5 s (horizontal). **(b)** Spikeforms. Scale bars, 50 μ V (vertical), 2 ms (horizontal). **c**, Color mapping of synchronization score between the electrodes located inside the 4-month-old cortical organoid. **d**, Neural networking map based on synchronization score and neural community of 4-month-old cortical organoid integrated on 3D LM MEA. **e**, Configuration of 3D LM electrodes with 3-level of heights. **f,g**, Representative intra-organoid signals depending on the heights of the electrodes. Each color represents the height of the electrodes; Orange, 50 μ m, green, 100 μ m, blue, 200 μ m. **(f)** SU potentials. Scale bars, 100 μ V (vertical), 0.5 s (horizontal). **(g)** Spikeforms. Scale bars, 100 μ V (vertical), 0.5 ms (horizontal). **h**, 3D neural networking map corresponding to the 3-level height variance of 3D LM MEA. **i,j**, Changes in neural activity depending on the heights of the electrodes ($n = 5$ electrodes per each height); **(i)** Spiking rate, **(j)** Number of bursts. All data are presented as mean \pm s.e.m.

Minor comment 7: "Page 31: There is a red font."

Response to Minor comment 7:

We have revised the red font.

Revised manuscript (page 36):

For the calculation of SPIKE-distance, we used Python package Pyspike (version 0.7.0) on a Python environment (version 3.9.2)⁴³⁴⁶.

[Response to Reviewer#2]

We appreciate your deliberate review of this manuscript with the thoughtful comments, and we welcome the opportunity to address and clarify the issues. Our response to the points raised in the comment is as follows:

Comment 1: “If the magnetic movement shifts the entire set of needles by a small amount in one direction, couldn’t the same result be obtained by pushing the organoids slightly? Please justify. The novelty of using magnetically driven deflection should be the ability to move the needles in other complex configurations.”

Response to Comment 1:

As the reviewer mentioned, slightly pushing the organoids enables the acquisition of neural signals from different spots, similar to that we change the recording sites within the brain organoid by the magnetically shifting the electrodes. However, pushing organoids does not provide precise control over their movement, which hinders detailed shifts in the recording site and poses a risk of causing detrimental damage. In contrast, we employ magnetic tilting of electrodes under the fine control of a magnet to effectively increase the recording site density while maintaining the cytoarchitecture of the organoids. Since detailed control of the magnet enables minute deflections of electrodes, we constructed the magnetic tilting system with a 6-axis stage that enables a minimum magnet displacement of $0.5\ \mu\text{m}$. To investigate the precise controllability of our magnetic tilting system, we theoretically studied the minimum degree of electrode (diameter: $20\ \mu\text{m}$, height: $260\ \mu\text{m}$) deflection under our system, as shown in Supplementary Note 4. Our system allows the electrode to tilt with a resolution of 0.002° , resulting in a minimum displacement of electrode tip of $0.009\ \mu\text{m}$. When examining the tilting trajectory of an electrode (height: $260\ \mu\text{m}$, magnet speed: $5\ \text{mm/s}$) as shown in newly added Supplementary Figure 22, the tilting degrees of the electrode are linearly increased along the magnet displacement with a slope of 3.34, which represents 0.00334° of tilting per $1\ \mu\text{m}$ of magnet displacement. The displacements of electrode tip are also changed along the magnet displacement with a slope of 13.99, $0.0140\ \mu\text{m}$ of electrode tip displacement per $1\ \mu\text{m}$ of magnet displacement, which is corresponded to our theoretical study. This high degree of controllability over the electrodes leads to substantial expansion of the coverable area by a single electrode ($78.5\ \mu\text{m}^2$ to $6.994 \times 10^4\ \mu\text{m}^2$) as shown in Fig. 6f.

[**Supplementary Figure 22 (newly added).** Tilting behavior of 3D LM pillars along the magnet displacement. **a**, Change in tilting degree along the magnet displacement. The slope represents that the magnetically tiltable electrode deflects 3.34° for $1\ \text{mm}$ of magnet displacement. **b**, Change in displacement of electrode tip along the magnet displacement. The slope represents that the tip of magnetically tiltable electrode shifts $13.99\ \mu\text{m}$ for $1\ \text{mm}$ of magnet displacement.]

Furthermore, considering that organoids do not exhibit solid morphologies, physical pushing can induce deformation on the organoids, whereas our system preserves the intact structure of the organoids due to the softness of the liquid metal. To verify the absence of the damage on the brain organoids, we examined sectioned slices of organoids subjected to the magnetic tilting of electrodes (thickness of cobalt: $120\ \text{nm}$). After coating the 3D liquid metal (LM) multi-electrode array (MEA) with a cell-adhesive layer (laminin and poly-D-lysine), we integrated the organoids onto the array and performed 20 cycles of repetitive tilting by moving the magnet (distance from the MEA: $2\ \text{cm}$) at speed of $5\ \text{mm/s}$ for 4 seconds each cycle of

tilting. The movement of the organoid during the magnetic tilting was not observable as shown in newly added Supplementary Video 9. This result ensures the magnetic tilting of electrodes within the organoid while the organoid is securely immobilized onto the MEA. Then, to compare the damage of the organoid inserted with the magnetically tiltable 3D LM electrodes (height: 150 μm , thickness of cobalt layer: 120 nm), we performed 3D visualization of the organoids both without and with magnetic tilting. The electrodes were tilted repetitively for 20 cycles by moving the magnet (distance from the MEA: 2 cm) at speed of 5 mm/s for 4 seconds each cycle of tilting. The organoids with magnetic tilting were fixed with the 3D LM electrodes at the tilted position and stained with TUJ1, phosphorylated Tau (p-Tau), and DAPI. Whole-mount 3D staining analysis revealed that the 3D LM electrodes remained intactly inserted into the organoid in both conditions, and the overall architecture of the organoid was well preserved. Additionally, comparison of p-Tau expression, a marker of brain damage, confirmed that magnetic tilting did not significantly damage the organoid. This suggests that magnetic tilting of 3D LM electrode does not cause significant damage to the inserted organoid while altering the interaction of the electrodes with neural cells, potentially allowing for a broader recording of neural network activity within the organoid. We have included these results to demonstrate the fine control of magnetic tilting of electrodes within the organoid while maintaining the intact cytoarchitecture.

[**Supplementary Figure 28 (newly added)**. Whole-mount 3D imaging of optically cleared organoids without (left) and with (right) magnetic titling of 3D LM electrodes. For the organoid with magnetic tilting, the organoid was fixed with the 3D LM electrodes at the tilted position. The organoids were stained with TUJ1, phosphorylated Tau (p-Tau), and DAPI for investigation of internal architecture and damage of organoids.]

Revised manuscript (pages 18-19):

To investigate the precise controllability of our magnetic tilting system, we theoretically studied the minimum degree of electrode deflection under our system, as shown in Supplementary Note 4. Our system allows the electrode to tilt with a resolution of 0.002° , resulting in a minimum displacement of electrode tip of $0.009 \mu\text{m}$. When examining the tilting trajectory of an electrode (height: $260 \mu\text{m}$, magnet speed: 5 mm/s) as shown in Supplementary Fig. 22, the tilting degrees of the electrode are linearly increased along the magnet displacement with a slope of 3.34, which represents 0.00334° of tilting per $1 \mu\text{m}$ of magnet displacement. The displacements of electrode tip are also changed along the magnet displacement with a slope of 13.99, $0.0140 \mu\text{m}$ of electrode tip displacement per $1 \mu\text{m}$ of magnet displacement, which is corresponded to our theoretical study.

Revised manuscript (page 22):

As the brain organoids are securely immobilized onto the substrate of 3D LM MEAs with the aid of cell-adhesive coating, the brain organoids do not shift under the magnetic tilting of electrodes within them (Supplementary Video 9).

Revised manuscript (pages 22-23):

In addition, the 3D visualization of the organoids both without and with magnetic tilting (electrodes at the tilted position) was performed to examine the damage that might be induced due to the 20 cycles of magnetic tilting (magnet speed of 5 mm/s for 4 seconds per each tilting). As shown in Supplementary Fig. 28, the magnetically tiltable electrodes remained intactly and the cytoarchitecture of the organoid was well preserved even after the 20 cycles of tilting. Additionally, a comparison of phosphorylated Tau (p-Tau) expression, a marker of brain damage, confirms that the magnetic tilting of 3D LM electrodes within the organoid does not cause any damage to the inserted organoid while

altering the interaction of the electrodes with neural cells, potentially allowing for a broader recording of neural network activity within the organoid.

Revised Supplementary Information (page 25):

Supplementary Figure 22. Tilting behavior of 3D LM pillars (height: 260 μm) along the magnet displacement. **a**, Change in tilting degree along the magnet displacement. The slope represents that the magnetically tiltable electrode deflects 3.34 $^\circ$ for 1 mm of magnet displacement. **b**, Change in displacement of electrode tip along the magnet displacement. The slope represents that the tip of magnetically tiltable electrode shifts 13.99 μm for 1 mm of magnet displacement.

Revised Supplementary Information (page 31):

Supplementary Figure 28. Whole-mount 3D imaging of optically cleared organoids without (left) and with (right) magnetic titling of 3D LM electrodes. For the organoid with magnetic tilting, the organoid was fixed with the 3D LM electrodes at the tilted position. The organoids were stained with TUJ1, phosphorylated Tau (p-Tau), and DAPI for investigation of internal architecture and damage of organoids.

Revised Supplementary Information (page 41):

Supplementary Video 9. Immobilization of brain organoid under magnetic tilting of electrode.

Revised Supplementary Information (pages 46-47):

Supplementary Note 4 | Precise controllability of our magnetic tilting system

To investigate the precise controllability of our magnetic tilting system, we calculate the minimum tilting degree and the minimum displacement of electrode tip when the magnet moves 0.5 μm , which is the minimum magnet displacement under our magnetic tilting system. Since the bottom of the magnetically tiltable electrodes are fixed on the substrate, the tilting degree of the electrode tip depends on the torque force exerted by the magnetic field, as described by the equation,

$$\theta = \frac{\tau \cdot L}{E \cdot I}$$

where θ is the tilting degree of magnetically tiltable electrode, τ is the torque force exerted on the electrode by the magnetic field gradient, L is the height of the electrode, E is the elastic modulus of the electrode, and I is the area moment of inertia. As the magnetically tiltable electrodes are forced to be

tilted by the magnetic layer on the surface under the magnetic field gradient, τ is described by the following equation,

$$\tau = L \cdot F \cdot N_{Co} = L \cdot m_{Co} \nabla B \cdot N_{Co} \approx L \cdot m_{Co} \cdot \frac{3B}{r} \cdot N_{Co}$$

where F is the exerted force on the electrode, N_{Co} is the number of cobalt atom of the electrode, m_{Co} is the magnetic moment of cobalt, which is $1.72\mu_B$ (*Bohr magneton*; $\mu_B = 9.274 \times 10^{-24} A \cdot m^2$), ∇B is the magnetic field gradient, B is the magnetic flux density, and r is the displacement of the magnet. The magnet of our magnetic tilting system exhibits the magnetic flux density of 285 mT, and the magnetic flux density at the distance of 2 cm from the magnet is 4.45 mT since the magnetic field decreases by the 3 orders of magnitude. Furthermore, considering the magnetically tiltable electrode (diameter: 20 μm , height: 260 μm) is covered by 120 nm thickness of cobalt (density: 8.9 g/cm³) on the half side, N_{Co} is 8.913×10^{13} atom. As a result, τ is calculated as 9.872×10^{-9} Nm.

As the magnetically tiltable electrodes are composed of EGaIn and cobalt, the elastic modulus of the electrodes is calculated considering the volume percentages of each material, which is 98.8% and 1.2%, respectively. Considering the elastic modulus of each material (210 GPa for cobalt and 210 kPa for EGaIn), the elastic modulus of the magnetically tiltable electrode is 2.53 GPa. Consequently, the minimum tilting degree of the magnetically tiltable electrode is calculated as 0.002 ° and the minimum displacement of electrode tip is 0.009 μm .

Comment 2: “Some studies talk about the effect of magnetic fields on changes in neural activity and ion channels, so it might be good to show a study (besides the change in the noise level that the authors presented) where the firing before and after the magnet is introduced.”

Response to Comment 2:

We thank the reviewer for the insightful comment. As the reviewer suggested, we analyzed and compared the spiking rates of single-unit potentials with and without an external magnetic field (newly added Supplementary Figure 32). For accurate analysis on the effect of magnetic field on firing rate by excluding any possible deformation of electrodes, we loaded brain organoid on 3D LM MEA with non-magnetic stationary electrodes. Significant changes in spiking rate along the existence of the magnetic field generated by the magnet were not observable, indicating that the magnetic field did not affect on the spontaneous neural activity of the organoids. To induce change in neural activity by magnetic field, strong magnetic field whose strength is over 3 T should be exerted on the neural tissues [*Neuron*, 55, 187, 2007]. However, since the strength of magnet utilized in our system is 285 mT on the surface, it cannot induce any change in neural activity of organoids while deflecting magnetic tiltable electrodes.

[**Supplementary Figure 32 (newly added)**]. Spontaneous neural activities of brain organoids detected by 3D LM MEA without and with external magnetic field. **a**, Single-unit potentials from a single brain organoid recorded by 3D LM MEA without (black) and with (red) external magnetic field ($n =$ for 3 electrodes per each group). Scale bars, 50 μV (vertical), 0.5 s (horizontal). **b**, Comparison of spiking rates

between single-unit potentials recorded without and with external magnetic field. Statistical differences were determined with unpaired, one-sided t-test; n.s. indicates not significant.]

Revised manuscript (page 23):

For accurate analysis on the effect of magnetic field on firing rate by excluding any possible deformation of electrodes, we loaded brain organoid on 3D LM MEA with non-magnetic stationary electrodes. Significant changes in spiking rate along the existence of the magnetic field generated by the magnet were not observable, indicating that the magnetic field did not affect on the spontaneous neural activity of the organoids since the strength of the magnet of our system (285 mT on the surface) is not large enough to induce any changes in neural activities while deflecting magnetic tiltable electrodes (Supplementary Fig. 32).

Revised Supplementary Information (page 35):

Supplementary Figure 32. Spontaneous neural activities of brain organoids detected by 3D LM MEA without and with external magnetic field. **a**, Single-unit potentials from a single brain organoid recorded by 3D LM MEA without (black) and with (red) external magnetic field ($n = 3$ electrodes per each group). Scale bars, 50 μ V (vertical), 0.5 s (horizontal). **b**, Comparison of spiking rates between single-unit potentials recorded without and with external magnetic field. Statistical differences were determined with unpaired, one-sided t-test; n.s. indicates not significant.

Comment 3: "It would be good to analyze the power spectral density of the LFP data further to see any peculiar effects."

Response to Comment 3:

As the reviewer suggested, we analyzed the power spectral density of LFP recorded during the magnetic tilting of the electrodes. No significant abnormalities during the magnetic tilting were observed, indicating that magnetic field does not affect the spontaneous neural activity of the organoids. This result has been included as Supplementary Figure 33b.

[**Supplementary Figure 33b (newly added)**. Power spectral density analysis during the magnetic tilting of electrodes.]

Revised manuscript (page 23):

Furthermore, Fig. 6g, h and Supplementary Fig. 4733 demonstrate that a single electrode in MEA can collect electrophysiological signals from multiple points within a cortical organoid, effectively densifying the recording sites without needing additional electrodes.

Revised Supplementary Information (page 36):

Supplementary Figure 4733. a, Local field potential recorded by magnetically reshapable 3D LM MEA. Scale bars, 500 μV (vertical), 5 s (horizontal). **b**, Power spectral density analysis during the magnetic tilting of electrodes.

Comment 4: “Based on the fabrication steps provided, could the researchers fabricate magnetic needles first and then nonmagnetic on the same chip to simultaneously record using both needles to see how the magnetic field affects unit detection? If that experiment results in no changes in nonmagnetic electrodes and detection of different units before and after deflection in magnetic electrodes, that would be more conclusive.”

Response to Comment 4:

To fabricate the 3D LM MEA that includes both non-magnetic and magnetic electrodes, we first printed 3D electrodes (height: 250 μm) on 8 of the 16 electrode pads. The electrode pads where the non-magnetic electrodes would be printed were covered, and we used the shadow mask, as shown in Supplementary Figure 9, to selectively deposit cobalt onto the printed electrodes. After the cobalt layer was deposited, the remaining 8 electrodes were printed at the same height of 250 μm and subsequently passivated with a 2 μm -parlylene layer. The post-fabrication process and organoid integration followed the procedure described in Methods section. Then, the 3D LM MEA was placed on the recording interface of the magnetic tilting system, and neural signals were recorded during the magnet movement (speed: 5 mm/s, displacement from the MEA: 2 cm) for magnetic tilting. Based on the single-unit potentials detected by both non-magnetic and magnetic electrodes, we analyzed the spiking rates, spikeforms, and spike waveform similarity to assess the influence of the magnetic field on neural signal detection. The representative single-unit potentials and spikeforms have been included in Supplementary Figure 34.

[**Supplementary Figure 34 (newly added)**. Neural activities (single-unit potentials, raster plot, and spikeforms) detected by both non-magnetic stationary electrodes and magnetic electrodes from a single organoid. Each row presents representative single unit potential (top), raster plot (middle), and spikeforms (bottom) from a single electrode. The left column shows the neural activities before the magnetic tilting and the right column shows them after the magnetic tilting. Scale bars for single-unit potentials, 50 μ V (vertical), 0.5 s (horizontal). Scale bars for spikeforms, 20 μ V (vertical), 1 ms (horizontal).]

As shown in Supplementary Figure 35a, the spiking rates of both non-magnetic electrodes and magnetic electrodes do not show specific tendency in response to the magnetic tilting, indicating that the spiking rates reflect neural activities of brain organoids regardless of the magnetic tilting. For a more detailed investigation of spikeform changes, we calculate the spike waveform similarity between spikes from the same electrode before and after the magnetic tilting (Supplementary Figure 35b). Negligible changes are observed in the spikeforms of neural signals from non-magnetic electrodes before and after the magnetic tilting as further verified by spike waveform similarity of 0.911, suggesting that the non-magnetic electrodes consistently detected the same group of neurons, unaffected by magnetic tilting. In contrast, the spikeforms detected by the magnetic electrodes differ after the magnetic tilting, showing the spike waveform similarity of 0.740, which indicates detection of different group of neurons.

[**Supplementary Figure 35 (newly added)**. Comparison of neural activities between non-magnetic electrodes and magnetically tiltable electrodes. **a**, Changes in spiking rates before and after the magnetic tilting. **b**, Changes in spike waveform similarities between before and after the magnetic tilting for each electrode ($p = 0.02353$). The similarities of spike waveforms of magnetically tiltable electrodes show lower value than that of non-magnetic stationary electrodes. This result indicates that the magnetically tiltable electrodes effectively shift the recording spots within the organoid to detect the other group of neurons, whereas the non-magnetic stationary electrodes consistently detect the same group of neurons, unaffected by the magnetic tilting. All data are presented as mean \pm s.e.m. and statistical differences were determined with unpaired, one-sided t -test; $*p < 0.05$.]

These results present that the non-magnetic electrodes are not influenced by the magnetic field gradient generated by magnet movement, while magnetic electrodes shift their recording spots through the magnetic tilting. By simultaneously monitoring the neural signal changes from both non-magnetic and magnetic electrodes within the same organoid along with magnet movement, we reliably confirmed that the magnetically tiltable electrodes can detect different groups of neurons by shifting the recording site without introducing additional changes related to the presence of magnetic field. We have supplemented these data as Supplementary Figures 34, 35 and descriptions in revised manuscript to further verify the multi-spot detection by our magnetically tiltable 3D LM electrodes.

Revised manuscript (page 24):

To further verify the multi-spot detection by magnetically tilted 3D LM electrodes, we fabricated the non-magnetic stationary electrodes and magnetically tiltable electrodes in a single substrate and recorded neural signals by them from a single organoid (Supplementary Fig. 34). The spiking rates of both non-magnetic electrodes and magnetic electrodes did not show specific tendency in response to the magnetic tilting, indicating that the spiking rates reflect neural activities of brain organoids

regardless of the magnetic tilting (Supplementary Fig. 35a). On the other hand, the spike waveform similarity of magnetically tiltable electrodes showed much lower value (0.740) compared to that of non-magnetic stationary electrodes (0.911) (Supplementary Fig. 35b), showing the detection of different group of neurons by magnetic tilting. These results indicate that the non-magnetic electrodes are not influenced by the magnetic field gradient generated by magnet movement, while magnetic electrodes shift their recording spots through the magnetic tilting.

Revised Supplementary Information (pages 37-38):

Supplementary Figure 34. Neural activities (single-unit potentials, raster plot, and spikeforms) detected by both non-magnetic stationary electrodes and magnetic electrodes from a single organoid. Each row presents representative single unit potential (top), raster plot (middle), and spikeforms (bottom) from a single electrode. The left column shows the neural activities before the magnetic tilting and the right column shows them after the magnetic tilting. Scale bars for single-unit potentials, 50 μV (vertical), 0.5 s (horizontal). Scale bars for spikeforms, 20 μV (vertical), 1 ms (horizontal).

Revised Supplementary Information (page 39):

Supplementary Figure 35. Comparison of neural activities between non-magnetic electrodes and magnetically tiltable electrodes. **a**, Changes in spiking rates before and after the magnetic tilting. **b**, Changes in spike waveform similarities between before and after the magnetic tilting for each electrode ($p = 0.02353$). The similarities of spike waveforms of magnetically tiltable electrodes show lower value than that of non-magnetic stationary electrodes. This result indicates that the magnetically tiltable electrodes effectively shift the recording spots within the organoid to detect the other group of neurons, whereas the non-magnetic stationary electrodes consistently detect the same group of neurons, unaffected by the magnetic tilting. All data are presented as mean \pm s.e.m. and statistical differences were determined with unpaired, one-sided t -test; $*p < 0.05$.

Comment 5: “Authors should discuss the potential shortcomings of the presented method. For instance, direct laser writing methods can make complex shapes, albeit at a slower throughput. Also, the current designs can only do inter-organoid recording from the bottom half of the organoid. Would it be possible to print curved needles to collect data from the top half of the organoids (similar to the interconnects the authors presented in their earlier studies)?”

Response to Comment 5:

We present a 3D liquid metal-electrode array for electrophysiological monitoring of brain organoids, which facilitates magnetic tilting and minimizes damage to the brain organoids. Our approach involves direct printing of liquid metal to form 3D pillar structure of electrodes. This method allows for the fabrication of 3D soft electrodes of various heights at desired positions, enabling the customization of 3D MEA tailored to the shape and size of the organoid. Nonetheless, several potential shortcomings should be addressed to expand the applicability of this technique. Firstly, the printing system uses a single nozzle, making it time-consuming for mass production. Additionally, since the liquid metal is printed under the ambient conditions, this method has structural constraints compared to other printing methods, such as direct laser writing or inkjet printing, which utilize photocurable, thermoset, or rigid materials. While the liquidous properties of EGaIn present these challenges, it also offers irreplaceable advantages including minimized tissue damage and robust magnetic tilting for high-resolution electrophysiological analysis of brain organoids. We have included this content regarding the potential shortcomings of presented method in the discussion section of the manuscript.

Printing curved needles on the top of the organoid, which is suggested by the reviewer, is not feasible since the brain organoids are cultivated in the media. The process of printing liquid metal into a 3D pillar structure requires strong adhesion to the substrate. However, in an aqueous condition, the high surface tension of the liquid metal prevents proper wetting on the substrate, resulting in the failure to form 3D pillar

structure. As an alternative, we propose a new structure to collect data from the top half of the organoid. As the softness of the liquid metals enables to conform to the 3D curvature of the organoid, we designed a 3D liquid metal electrode array on the Parylene-C substrate that can attach to the surface of top half of the organoid. We printed the interconnection lines and the 3D electrodes (height: 200 μm) using liquid metals on a Parylene-C (thickness: 1 μm) deposited on water-soluble polyethylene oxide (PEO) film. Then, 2 μm of Parylene-C was deposited for the passivation and the electrode tips were selectively opened for neural recording. After the organoid was integrated onto the 3D LM MEA for bottom half recording, we flipped the film to insert 3D LM electrodes for top half recording onto the brain organoid. Owing to the deformability of the liquid metal, the film seamlessly covered the 3D spherical organoid as shown in Figure R1. We believe that the intrinsic softness of liquid metals offers unconstrained flexibility of designing electrodes which can be applied to various subjects including brain organoids.

[**Figure R1.** 3D stereomicrograph of brain organoid integrated onto the 3D LM MEA for bottom half recording and covered by Parylene film with 3D LM electrodes for top half recording.]

Revised manuscript (page 26):

Nonetheless, several potential shortcomings should be addressed to expand the applicability of this technique. Firstly, the printing system uses a single nozzle, making it time-consuming for mass production. Additionally, since the liquid metal is printed under the ambient conditions, this method has structural constraints compared to other printing methods, such as direct laser writing or inkjet printing, which utilize photocurable, thermoset, or rigid materials. While the liquidous properties of EGaIn present these challenges, it also offers irreplaceable advantages including minimized tissue damage and robust magnetic tilting for high-resolution electrophysiological analysis of brain organoids.

Nevertheless, Furthermore, we note that challenges still remain, given the need for the development of reliable brain organoids that resemble the full functionality of the human brain.

Comment 6: “Organoid placement strategy – Media is aspirated after placing the organoid on top of the needle array to “settle down” the organoid. Authors claim that the electrodes are “naturally” inserted into the organoid during this process. It is unclear how much of the media is aspirated and whether capillary forces drive the insertion or the gravity. If it is the capillary, does it affect the needles themselves (elastocapillary effect) since they are soft?”

Response to Comment 6:

For better understanding of the organoid insertion process into the 3D LM MEA, we recorded the procedure as shown in newly added Supplementary Video 2. Initially, the organoid is placed into the 3D LM MEA device, which contains culture media. At this stage, the 3D LM electrodes are densely distributed, and the organoid is light enough to rest gently on top of the electrode array without crushing them. Then, the media is gradually aspirated using a micropipette (aspirated media volume: 1000 μl), thereby eliminating the buoyancy from the media. This leaves gravity as the dominant acting on the organoid. As the organoid settles due to gravity, the organoid adheres to the bottom of the device, with the electrodes naturally inserting into the organoid during this process. Once the media is replenished, the organoid remains securely attached to the device, even when shaken. After subsequent culturing, the organoid becomes firmly attached to both the electrodes and the bottom of the device, which is coated with

poly-L-lysine (PLL) and laminin. We have newly added the video (Supplementary Video 2) showing the robust attachment of brain organoids onto the device.

Furthermore, to demonstrate the insertion of 3D LM electrodes into the brain organoids, we performed optical clearing of the organoid followed by whole-mount 3D imaging (newly added Supplementary Figure 6). The organoids were integrated onto the 3D LM MEA (electrode height: 150 μm) and stained with TUJ1 and DAPI for further investigation of internal architecture of the organoid, which could potentially be deformed by the inserted electrodes.

[**Supplementary Figure 6 (newly added)**. Whole-mount 3D imaging of optically cleared organoids without (left) and with (right) inserted 3D LM electrodes. The organoids were stained with TUJ1 and DAPI for investigation of internal architecture of organoids.]

By analyzing the entire region where the electrodes were inserted, we confirmed that the 3D LM electrodes were inserted into the organoid straight, maintaining their structure, which ensures the robust structural stability of 3D LM electrodes during the insertion process. The organoid also preserved its characteristic rosette structure, which is a specific structure of brain organoids, indicating the absence of structural damage to the organoid. This suggests that the 3D LM electrodes can be stably inserted into the organoid without significantly compromising their structural integrity, allowing for reliable monitoring of neural network within the organoid through the inserted 3D LM electrodes. We have supplemented these results as Supplementary Figure 6 and explanation in the revised manuscript to reinforce the reliability of our insertion process.

Revised manuscript (page 11):

Fig. 3a and Supplementary Video 2 illustrates the integration of this cortical organoid to our 3D LM MEA.

Revised manuscript (pages 11):

Furthermore, we performed optical clearing of the organoid followed by whole-mount 3D imaging (Supplementary Fig. 6), showing that the 3D LM electrodes were inserted into the organoid straight without causing any structural damage on brain organoids, allowing for reliable monitoring of neural network within the organoid.

Revised manuscript (page 33):

3D imaging of the cleared cortical organoids

3D imaging of the cleared cortical organoids was performed using a slightly modified protocol from previous studies^{48,49}. The organoids were fixed in 10% formalin solution (#HT501640, Sigma) for two hours at room temperature and washed in phosphate-buffered saline (PBS, Biosesang, Seongnam, Korea). The fixed organoids were washed with a washing buffer and treated with a blocking buffer for 8 h at 37 °C. Next, the organoids were incubated with primary antibodies for 67 h at 37 °C. After washing with the washing buffer, the organoids were incubated with secondary antibodies and DAPI solution for 68 h at 37 °C. Following this, the organoids were washed again with the washing buffer and cleared by replacing the Ce3D solution overnight. Images were acquired using a confocal microscope (LSM 700 and 880, Carl Zeiss) and processed with ImageJ (National Institutes of Health).

Revised Supplementary Information (page 8):

Supplementary Figure 6. Whole-mount 3D imaging of optically cleared organoids without (left) and with (right) inserted 3D LM electrodes. The organoids were stained with TUJ1 and DAPI for investigation of internal architecture of organoids.

Revised Supplementary Information (page 41):
Supplementary Video 2. Organoid insertion process.

***Comment 7:** “While the increase in firing rate following the KCl introduction could indicate that the activity is from organoids, having a negative control by using neurotoxins like TTX could further confirm that the captured data is not noise. Especially since the reported noise level (13 microvolts) is not particularly low.”*

Response to Comment 7:

As the reviewer suggested, we observed the response of the brain organoid to the treatment of 1 μM tetrodotoxin (TTX). The neural activities of the brain organoid effectively decreased after the treatment of TTX, which were successfully captured by our 3D LM MEA. We have added the results in Supplementary Figure 9 to verify the reliability of recording signals by 3D LM MEAs.

[**Supplementary Figure 9 (newly added).** Electrophysiological responses of cortical organoids to tetrodotoxin (TTX) treatment. **a**, Single unit potentials and raster plots recorded by 5 individual electrodes. Scale bars, 50 μV (vertical), 0.5 s (horizontal). **b**, Comparison in spiking rates of 5 individual electrodes before and after 1 μM TTX treatment ($p = 1.857 \times 10^{-8}$). All data are presented as mean \pm s.e.m. and statistical differences were determined with unpaired, one-sided t -test; $**p < 0.001$.]

Revised manuscript (page 13):

Along with the treatment of KCl, we examined the neural response of brain organoid to tetrodotoxin (TTX) to confirm that the recorded signals originated from neuronal activities (Supplementary Fig. 9).

Revised Supplementary Information (page 11):

Supplementary Figure 9. Electrophysiological responses of cortical organoids to tetrodotoxin (TTX) treatment. **a**, Single unit potentials and raster plots recorded by 5 individual electrodes. Scale bars, 50 μ V (vertical), 0.5 s (horizontal). **b**, Comparison in spiking rates of 5 individual electrodes before and after 1 μ M TTX treatment ($p = 1.857 \times 10^{-8}$). All data are presented as mean \pm s.e.m. and statistical differences were determined with unpaired, one-sided *t*-test; ** $p < 0.001$.

[Additional Modification]

Modification 1:

The authors have been newly added.

Revised manuscript (page 1):

Enji Kim^{1,4,5†}, Eunseon Jeong^{2†}, **Yeon-Mi Hong^{1,4†}**, Inhea Jeong^{1,4,5}, Junghoon Kim^{2,4}, Yong Won Kwon^{1,4,5}, Young-Geun Park^{1,4,5}, **Jiin Lee²**, Suah Choi², Ju-Young Kim^{4,5}, Jae-Hyun Lee^{4,5★}, Seung-Woo Cho^{2,4,5★}, Jang-Ung Park^{1,3,4,5,6★}.

Revised manuscript (page 1):

¹Department of Materials Science and Engineering, Yonsei University, Seoul 03722, Republic of Korea

²Department of Biotechnology, Yonsei University, Seoul 03722, Republic of Korea

³Department of Neurosurgery, Yonsei University College of Medicine

⁴Center for Nanomedicine, Institute for Basic Science (IBS), Yonsei University, Seoul, 03722, Republic of Korea.

⁵Graduate Program of Nano Biomedical Engineering (NanoBME), Advanced Science Institute, Yonsei University

⁶Yonsei-KIST Convergence Research Institute, Seoul 03722, Republic of Korea

Revised manuscript (page 42):

Author contributions

E. K. and E. J. carried out the experiment, analyzed the data, and wrote the manuscript. **E. K., Y.-M. H.**, I. J., Y. W. K., Y.-G. P., and J.-Y. K were involved in device fabrications and electrophysiological analysis. **E. J., J. K., J. L** and S. C. were involved in organoid cultivation and assessment of biocompatibility. J.-U. P., S.-W. C., and J.-H, L an oversaw all of the research phases and revised the manuscript. All authors discussed and commented on the manuscript.

Revised Supplementary Information (page 1):

Enji Kim^{1,4,5†}, Eunseon Jeong^{2†}, **Yeon-Mi Hong^{1,4†}**, Inhea Jeong^{1,4,5}, Junghoon Kim^{2,4}, Yong Won Kwon^{1,4,5}, Young-Geun Park^{1,4,5}, **Jiin Lee²**, Suah Choi², Ju-Young Kim^{4,5}, Jae-Hyun Lee^{4,5★}, Seung-Woo Cho^{2,4,5★}, Jang-Ung Park^{1,3,4,5,6★}.

Revised Supplementary Information (page 1):

¹Department of Materials Science and Engineering, Yonsei University, Seoul 03722, Republic of Korea

²Department of Biotechnology, Yonsei University, Seoul 03722, Republic of Korea

³Department of Neurosurgery, Yonsei University College of Medicine

⁴Center for Nanomedicine, Institute for Basic Science (IBS), Yonsei University, Seoul, 03722, Republic of Korea.

⁵Graduate Program of Nano Biomedical Engineering (NanoBME), Advanced Science Institute, Yonsei University

⁶Yonsei-KIST Convergence Research Institute, Seoul 03722, Republic of Korea

Modification 2:

Acknowledgement has been revised.

Revised manuscript (page 42):

This work was supported by the Ministry of Science & ICT (MSIT), the Ministry of Trade, Industry and Energy (MOTIE), the Ministry of Health & Welfare, and the Ministry of Food and Drug Safety of Korea through the National Research Foundation (~~2021R1A2C3004262, 2023R1A2C2006257, RS-2024-00430782RS-2024-00460364~~), and ERC Program (~~2022R1A5A6000846RS-2024-00406240~~), ~~and the Korea Medical Device Development Fund grant (RMS-2022-11-1209 / KMDF RS-2022-00141392). This work was also supported by Institute for Basic Science (IBS-R026-D1).~~

Modification 3:

References have been changed and supplemented.

Revised manuscript (pages 38-41):

- ~~19. Fair, S. R. et al. Electrophysiological Maturation of Cerebral Organoids Correlates with Dynamic Morphological and Cellular Development. *Stem Cell Rep.* **15**, 855–868 (2020).~~
19. Fair, S. R. et al. Electrophysiological Maturation of Cerebral Organoids Correlates with Dynamic Morphological and Cellular Development. *Stem Cell Rep.* **15**, 855–868 (2020).
2019. Park, Y. et al. Three-dimensional, multifunctional neural interfaces for cortical spheroids and engineered assembloids. *Sci. Adv.* **7**, eabf9153 (2021).
2120. Huang, Q. et al. Shell microelectrode arrays (MEAs) for brain organoids. *Sci. Adv.* **8**, eabq5031 (2022).
21. Kim, J. et al. Intraocular pressure monitoring following islet transplantation to the anterior chamber of the eye. *Nano Lett.* **20**, 1517–1525 (2019).
22. Jang, J. et al. Human-interactive, active-matrix displays for visualization of tactile pressures. *Adv. Mater. Technol.* **4**, 1900082 (2019).
23. Park, J. et al. Research on flexible display at Ulsan National Institute of Science and Technology. *npj Flex. Electron.* **1**, 9 (2017).
24. Oh, S.-J. et al. Newly Designed Cu/Cu₁₀Sn₃ Core/Shell Nanoparticles for Liquid Phase-Photonic Sintered Copper Electrodes: Large-Area, Low-Cost Transparent Flexible Electronics. *Chem. Mater.* **28**, 4714–4723 (2016).
2225. Quadrato, G. et al. Cell diversity and network dynamics in photosensitive human brain organoids. *Nature* **545**, 48–53 (2017).
2326. Shin, H. et al. 3D high-density microelectrode array with optical stimulation and drug delivery for investigating neural circuit dynamics. *Nat. Commun.* **12**, 492 (2021).
2427. Polikov, V. S., Tresco, P. A. & Reichert, W. M. Response of brain tissue to chronically implanted neural electrodes. *J. Neurosci. Methods* **148**, 1–18 (2005).
- ~~2528. Lai, Jesse D., et al. A model of traumatic brain injury using human iPSC-derived cortical brain organoids. Preprint at <https://www.biorxiv.org/content/10.1101/2020.07.05.180299v1> (2022).~~
- Lai, Jesse D., et al. KCNJ2 inhibition mitigates mechanical injury in a human brain organoid model of traumatic brain injury. *Cell Stem Cell* **31**, 519–536 (2024).
2629. Le Floch, P. et al. Stretchable Mesh Nanoelectronics for 3D Single-Cell Chronic Electrophysiology from Developing Brain Organoids. *Adv. Mater.* **34**, 2106829 (2022).
2730. Park, Y.-G. et al. High-Resolution 3D Printing for Electronics. *Adv. Sci.* **9**, 2104623 (2022).
2831. Park, Y.-G. et al. Liquid metal-based soft electronics for wearable healthcare. *Adv. Healthc. Mater.* **10**, 2002280 (2021).
2932. Kim, M. et al. Multimodal Characterization of Cardiac Organoids Using Integrations of Pressure-Sensitive Transistor Arrays with Three-Dimensional Liquid Metal Electrodes. *Nano Lett.* **22**, 7892–7901 (2022).
3033. Yoon, S.-J. et al. Reliability of human cortical organoid generation. *Nat. Methods* **16**, 75–78 (2019).
3134. Zourray, C., Kurian, M. A., Barral, S. & Lignani, G. Electrophysiological Properties of Human Cortical Organoids: Current State of the Art and Future Directions. *Front. Mol. Neurosci.* **15**, 839366 (2022).
3235. Kreuz, T., Chicharro, D., Houghton, C., Andrzejak, R. G. & Mormann, F. Monitoring spike train synchrony. *J. Neurophysiol.* **109**, 1457–1472 (2013).
3336. Blondel, V. D., Guillaume, J.-L., Lambiotte, R. & Lefebvre, E. Fast unfolding of communities in large networks. *J. Stat. Mech. Theory Exp.* **2008**, P10008 (2008).
3437. Lam, D. et al. Tissue-specific extracellular matrix accelerates the formation of neural networks and communities in a neuron-glia co-culture on a multi-electrode array. *Sci. Rep.* **9**, 4159 (2019).
3538. Harb, K., et al. Area-specific development of distinct projection neuron subclasses is regulated by postnatal epigenetic modifications. *Elife* **5**, e09531 (2016).
3639. Livesey, M. R. et al. Maturation and electrophysiological properties of human pluripotent stem cell-derived oligodendrocytes. *Stem Cells* **34**, 1040–1053 (2016).
3740. Liu, Y. et al. Ferromagnetic Flexible Electronics for Brain-Wide Selective Neural Recording. *Adv. Mater.* **35**, 2208251 (2023).
3841. Jiang, S. et al. Spatially expandable fiber-based probes as a multifunctional deep brain interface. *Nat. Commun.* **11**, 6115 (2020).
3942. Schoonover, C. E., Ohashi, S. N., Axel, R. & Fink, A. J. P. Representational drift in primary olfactory cortex. *Nature* **594**, 541–546 (2021).

4043. Jang, J. *et al.* Disease-specific induced pluripotent stem cells: a platform for human disease modeling and drug discovery. *Exp. Mol. Med.* **44**, 202–213 (2012).
4144. Takahashi, K. *et al.* Induction of Pluripotent Stem Cells from Adult Human Fibroblasts by Defined Factors. *Cell* **131**, 861–872 (2007).
4245. Paşca, A. M. *et al.* Functional cortical neurons and astrocytes from human pluripotent stem cells in 3D culture. *Nat. Methods* **12**, 671–678 (2015).
4346. Mulansky, M., & Kreuz, T. PySpike—A Python library for analyzing spike train synchrony. *SoftwareX*, **5**, 183-189 (2016).
4447. Newman, M. E. J. & Girvan, M. Finding and evaluating community structure in networks. *Phys. Rev. E* **69**, 026113 (2004).
48. Li, W., Germain, R. N., & Gerner, M. Y. Multiplex, quantitative cellular analysis in large tissue volumes with clearing-enhanced 3D microscopy (Ce3D). *PNAS* **114**, E7321-E7330 (2017).
49. Lee, J. *et al.* Generation and characterization of hair-bearing skin organoids from human pluripotent stem cells. *Nature protocols* **17**, 1266-1305 (2022).

Modification 4:

There were errors regarding the statistical analysis in Figure 5. Furthermore, to provide accurate data for readers, we have revised Figure 5 and Supplementary Figure 19 by presenting box plots with the accurate data.

Revised manuscript (pages 54-56):

Fig 5. | Electrophysiological maturation of cortical organoid. **a**, Single unit potential of 1, 2, 4, 6-month-old cortical organoids recorded by the representative electrode located inside each organoid. Scale bars, 200 μ V (vertical), 2 s (horizontal). **b-e**, Changes in electrophysiological properties during maturation-span of cortical organoids ($n = 6$); **(b)** spiking rate, **(c)** interspike interval ($p = 0.01698$ for 2 and 4 months; $p = 0.02031$ for 2 and 6 months), **(d)** number of bursts, and **(e)** burst durations ($p = 0.03215$ for 2 and 6 months). **f**, Maturation of neural networking circuitry during maturation-span of cortical organoid. The error bars represent the s.e.m. Statistical differences were determined with unpaired, one-sided t -test; $*p < 0.05$. **g-j**, Changes in properties related to neural networking circuitry during maturation-span of cortical organoids ($n = 68$); **(g)** average number of connected lines per single node ($p = 0.01140$ for 2 and 6 months), **(h)** total number of connected lines within single organoid ($p = .009979$ for 2 and 4 months; $p = 9.715 \times 10^{-5}$ for 2 and 6 months), **(i)** number of nodes with connected lines within single organoid ($p = 8.823 \times 10^{-5}$ for 2 and 4 months; $p = 0.001193$ for 4 and 6 months; $p = 2.692 \times 10^{-10}$ for 2 and 6 months), **(j)** changes in properties related to neural community including number of neural communities ($p = 0.01165$ for 2 and 4 months; $p = 0.0002006$ for 2 and 6 months) and number of nodes not belong to the community within single organoid ($p = 5.704 \times 10^{-5}$ for 2 and 4 months; $p = 0.001194$ for 4 and 6 months; $p = 1.0610 \times 10^{-7}$ for 2 and 6 months). The error bars represent the s.e.m. Statistical differences were determined with unpaired, one-sided t -test; $*p < 0.05$, $**p < 0.01$, and $***p < 0.001$.

Revised Supplementary Information (page 22):

Supplementary Figure 819. Average number of nodes in a single community (left) and maximum number of nodes in a single community (right) during the maturation-span ($n = 8$). All data are presented as mean \pm s.e.m.

Modification 5:

We have modified Methods section with the revised information.

Revised manuscript (pages 29-30):

hiPSC maintenance

hiPSC lines ~~WT3 (kindly provided by the Yonsei University School of Medicine) and KYOU-DXR0109B (#ACS-1023; American Type Culture Collection, Manassas, VA, USA) were used~~ for cortical organoid generation^{40,44,44}. The use of hiPSCs was approved by the Institutional Review Board (IRB) of Yonsei University (Permit Number: ~~7001988-202104-BR-1167-01E~~7001988-202309-BR-1066-02E).

[**Response to Reviewer#1**]

We appreciate your thoughtful review of this manuscript, and we welcome the opportunity to address and clarify the valuable comments. Our response to the points raised in the comment is as follows:

Comment 1: “Supplementary Figure 18: What is the 3D morphology of the organoids after 7 weeks? Should the changes in morphology before and after 7 weeks be presented? Given that the authors coated the device with PDL-Laminin, which promotes neural cell adhesion and migration, is the 3D morphology of the organoids still intact after 7 weeks?”

Response to Comment 1:

We appreciate the reviewer for this valuable comment. When we imaged the electrode layer of the 3D poly-L-lysine (PLL)-laminin-coated MEA device integrated with brain organoid from day 1 to week 3, outgrowth of neuronal cells from the brain organoid was observed over culture time, which is attributed to PLL-laminin coating that promotes neural cell adhesion and migration as mentioned by the reviewer (Figure R1a,b). Nonetheless, overall structure and morphology of brain organoids could be maintained on the 3D liquid metal (LM) multi-electrode array (MEA) device. Whole-mount immunohistochemical staining for TUJ1 showing the 3-week-cultured brain organoid on the entire electrode layer of the 3D LM MEA device on also confirmed that cells extended only from the bottom region of the brain organoid in contact with the device, while the overall architecture of the brain organoid was maintained (Figure R1c,d). Despite outgrowth of neuronal cells, brain organoid was not disrupted and could retain its 3D intact morphology up to 7 weeks.

[**Figure R1.** Observation of brain organoid integrated with 3D liquid metal (LM) multi-electrode array (MEA) during long-term culture. **a**, Schematic illustration showing integration of brain organoid with the electrodes of the 3D LM MEA during long-term culture. **b**, Calcein AM-stained image of the bottom portion of brain organoid integrated with 3D LM MEA from day 1 to 3 weeks. **c,d**, Whole-mount 3D TUJ1 immunostaining image (c) and orthogonal image (d) of optically cleared brain organoid integrated with 3D LM MEA device for 3 weeks.]